# Numerical Solutions for Systems of Fractional and Classical Integro-Differential Equations via Finite Integration Method Based on Shifted Chebyshev Polynomials

**Ampol Duangpan** [1] , **Ratinan Boonklurb** [1,*] **and Matinee Juytai** [2]

1  Department of Mathematics and Computer Science, Faculty of Science, Chulalongkorn University, Bangkok 10330, Thailand; ampol.d@chula.ac.th
2  Winitsuksa School, Lopburi 15000, Thailand; 63010008@ws.ac.th
*  Correspondence: ratinan.b@chula.ac.th

**Abstract:** In this paper, the finite integration method and the operational matrix of fractional integration are implemented based on the shifted Chebyshev polynomial. They are utilized to devise two numerical procedures for solving the systems of fractional and classical integro-differential equations. The fractional derivatives are described in the Caputo sense. The devised procedure can be successfully applied to solve the stiff system of ODEs. To demonstrate the efficiency, accuracy and numerical convergence order of these procedures, several experimental examples are given. As a consequence, the numerical computations illustrate that our presented procedures achieve significant improvement in terms of accuracy with less computational cost.

**Keywords:** finite integration method; shifted Chebyshev polynomial; Caputo fractional derivative; system of fractional integro-differential equations; system of classical integro-differential equations

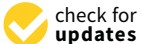



## 1. Introduction

Fractional calculus is a branch of mathematical analysis that has been receiving much attention from many researchers. Due to the fact that several real-world phenomena can be described successfully by developing mathematical models using fractional derivatives and integrations. Some interesting applications of fractional calculus can be found in various fields of sciences and engineering, for examples, viscoelasticity [1], nonlinear dynamical system [2], chaotic system [3], electromagnetic wave [4], heat transfer modeling [5], etc. In addition, the fractional calculus has grown in popularity as a tool for describing the physical features of real-world situations, particularly COVID-19, SIR model and health problems, as evidenced in [6,7] and references therein. One interesting issue regarding the fractional calculus is a fractional integro-differential equation (FIDE). It consists of both integral and differential operators involving derivatives of positive fractional order. The FIDEs have demonstrated to be adequate models for several phenomena arising in damping laws, earthquake model, diffusion processes, fluid dynamics, traffic models and acoustics, see [8–10] and references cited therein for more details. However, the fractional order derivative of FIDEs can be reduced to a positive integer order. Then, it is called the classical integro-differential equation (CIDE) which is frequently used to describe many applications which can be seen in [11–13] for details of applications. Actually, many problems of both FIDE and CIDE are often constructed to be a system.

In fact, most of FIDEs and CIDEs and system involving them are usually difficult to solve analytically. Therefore, numerical techniques are required to obtain an accurate approximate solution. Several numerical methods for solving the FIDEs and CIDEs have been given, for examples, variational iteration method [8], collocation method [13], homotopy method [14], Adomian's decomposition method [15]. In 2013, an efficient numerical

method had occurred which is called the finite integration method (FIM) introduced by
Wen et al. [16]. It has been developed in order to solve one-dimensional partial differential
equations (PDEs). The concept of FIM is to transform a given PDE into an equivalent inte-
gral equation and then numerical integrations are applied. It is known that the integration
task involves multiplication by a small step size, whereas the differentiation task involves
division by a small step size. As a reason, the numerical integration is very insensitive to
round-off error and preserves the approximation accuracy. Therefore, the approximation
of FIM can provide a stable, accurate and efficient numerical solution, see [16–18]. In 2015,
this FIM has been extended to overcome the multi-dimensional PDEs found in [17]. After
that, the FIM has been improved by hiring numerical quadratures such as Simpson's rule,
Newton Cotes and Lagrange interpolation, see [18]. As a consequence, these improved
FIMs give highly accurate solutions compared with the traditional FIM and finite difference
method (FDM). In 2018, Boonklurb et al. [19] have modified the FIM by using Chebyshev
polynomials to solve one- and two-dimensional PDEs. This modified FIM also provides
much higher accuracy than the FDM and those original FIMs with small computational
nodes. Recently, the modified FIM was widely utilized to apply with many applications,
see [20–23]. Also, it was demonstrated that results obtained by the modified FIM achieve
significant improvement in terms of accuracy more than several existing methods.

The major aim of this paper is to develop the modified FIM [19] by using the shifted
Chebyshev polynomial which thereafter will be referred to as FIM-SCP and also constructs
the operational matrix of fractional integration in order to devise two numerical procedures
for solving numerically the systems of both FIDEs and CIDEs of the Volterra type. Actually,
the technique of FIM-SCP has been proposed in [20]. It is used to find numerical solutions
of direct and inverse problems for the time-dependent Volterra integro-differential equation
(VIDE). Hence, in this paper, we continue our study from [20] by extending the VIDEs to
be a kind of system that involves the fractional-order differential operator. The problem
mainly considered in this article is called the system of FIDEs which is studied in the form
presented in [24], i.e.,

$$D^{\alpha_i} y_i(x) = f_i(x) + \sum_{j=1}^{m} \left( p_{ij}(x) y_j(x) + \int_0^x \kappa_{ij}(x,t) y_j(t) dt \right), \ \ x \in [0,L] \qquad (1)$$

for all $i \in \{1,2,3,\ldots,m\}$ and $L \in \mathbb{R}^+$ with initial conditions $y_i^{(v)}(0) = b_{v_i}$ for $v \in \{0,1,2,\ldots,\lceil \alpha_i \rceil - 1\}$, where $b_{v_i} \in \mathbb{R}$ is specified constant, $\alpha_i \in \mathbb{R}^+$ is parameter describing
the order of fractional derivative $D^{\alpha_i}$ in the Caputo sense [25], $\lceil \alpha_i \rceil$ is the smallest integer
greater than $\alpha_i$, $f_i(x)$ and $p_{ij}(x)$ are sufficiently continuous functions, $\kappa_{ij}(x,t)$ is continu-
ously integrable kernel function and $y_i(x)$ is unknown function to be solved numerically.
Next, when the derivative of fractional order is reduced to any $\alpha_i \in \mathbb{N}$, we obtain the system
of CIDEs. In this paper, we investigate the system of CIDEs in the following general form

$$\sum_{j=1}^{m} \mathcal{L}_{ij} y_j(x) = f_i(x) + \sum_{j=1}^{m} \lambda_{ij} \int_0^x \kappa_{ij}(x,t) y_j(t) dt, \ \ x \in [0,L] \qquad (2)$$

for all $i \in \{1,2,3,\ldots,m\}$ and $L \in \mathbb{R}^+$ with initial conditions $y_i^{(v)}(0) = b_{v_i}$ for $v \in \{0,1,2,\ldots,h_i - 1\}$, where $b_{v_i} \in \mathbb{R}$ is specified constant and $h_i = \max_{1 \le j \le m} r_{ij}$ when $r_{ij}$ is
the highest order of derivative for each $y_j$ contained in the linear differential operator $\mathcal{L}_{ij}$
which is defined by (33), $\lambda_{ij}$ is constant coefficient, $\kappa_{ij}(x,t)$ is continuously integrable kernel
function, $f_i(x)$ is continuous function and $y_j(x)$ is unknown function to be determined.
Moreover, we observe that if the kernel functions $\kappa_{ij}(x,t)$ or the constant coefficients $\lambda_{ij}$
in (2) are all zeros, (2) also becomes the system of ordinary differential equations (ODEs).
An interesting problem for the system of ODEs is the stiff problem which can be seen
in [26] for modeling various real-world problems. Nevertheless, the stiff system of ODEs
is difficult to solve analytically and numerically. A stiff system generally happens when
some components of the solutions decay much more rapidly than others. It affects their

numerical solutions in terms of stability. This feature forces the used numerical method to choose an extremely small step size which consumes the computational times expensively and may give inaccurate solutions. Accordingly, examples of the stiff system of ODEs are also presented to illustrate the efficiency of the proposed procedure that can be treated these troubles. In this study, we assume that (1) and (2) have unique solutions under the given supplementary conditions.

The organization of this paper is as follows. In Section 2, the developed FIM-SCP given by [20] is briefly introduced which is utilized to be the principal tool for devising the numerical procedures. In Section 3, the shifted Chebyshev expansion is employed to construct the operational matrix of fractional integration. It is together used with the FIM-SCP to devise the procedure for solving the system of FIDEs (1). This procedure is verified the efficiency with several examples. Next, Section 4 includes the procedure for solving the system of CIDEs (2) and experimental examples for testing accurate solutions obtained by the procedure. This procedure is also applied with the stiff system of ODEs via several examples in this section. Finally, the conclusion and discussion are summarized in Section 5.

## 2. The Developed FIM-SCP

In this section, we briefly introduce the technique of FIM-SCP presented in [20] which is utilized to be the principal tool for devising the numerical procedures to solve systems (1) and (2). Let us introduce the definition of shifted Chebyshev polynomials.

**Definition 1** ([27]). *The shifted Chebyshev polynomial of degree $n \geq 0$ is defined by*

$$S_n(x) = \cos\left(n \arccos\left(\frac{2x}{L} - 1\right)\right) \ \text{for} \ x \in [0, L]. \tag{3}$$

*Moreover, the analytic form of $S_n(x)$ with $n > 0$ given by [27] can be written as*

$$S_n(x) = n \sum_{k=0}^{n} (-1)^{n-k} \frac{(n+k-1)! \, 2^{2k}}{(n-k)!(2k)! \, L^k} x^k, \ \ x \in [0, L]. \tag{4}$$

Some important properties of the shifted Chebyshev polynomial are further given in Lemma 1. They will be used to construct the first and higher orders of the shifted Chebyshev integration matrices which are the major tools of the FIM-SCP.

**Lemma 1** ([20]). *The followings are properties of the shifted Chebyshev polynomial (3).*

*(i)   The zeros of shifted Chebyshev polynomial $S_n(x)$ for $x \in [0, L]$ are*

$$x_k = \frac{L}{2}\left(\cos\left(\frac{2k-1}{2n}\pi\right) + 1\right), \ k \in \{1, 2, 3, \ldots, n\}. \tag{5}$$

*(ii)   The $v$th order derivatives of shifted Chebyshev polynomial $S_n(x)$ at $x = 0$ are*

$$\frac{d^v}{dx^v} S_n(x)\Big|_{x=0} = (-1)^{v+n} \prod_{k=0}^{v-1} \frac{2}{L}\left(\frac{n^2 - k^2}{2k+1}\right). \tag{6}$$

*(iii)   The single integrations of shifted Chebyshev polynomial $S_n(x)$ for $n \geq 2$ are*

$$\int_0^x S_n(\xi)\, d\xi = \frac{L}{4}\left(\frac{S_{n+1}(x)}{n+1} - \frac{S_{n-1}(x)}{n-1} - \frac{2(-1)^n}{n^2 - 1}\right), \tag{7}$$

*where the initial integrations of $S_0$ and $S_1$ are $x$ and $\frac{x^2}{L} - x$, respectively.*

*(iv) The shifted Chebyshev matrix* **S** *at each node $x_k$ defined by (5) is*

$$\mathbf{S} = \begin{bmatrix} S_0(x_1) & S_1(x_1) & \cdots & S_{n-1}(x_1) \\ S_0(x_2) & S_1(x_2) & \cdots & S_{n-1}(x_2) \\ \vdots & \vdots & \ddots & \vdots \\ S_0(x_n) & S_1(x_n) & \cdots & S_{n-1}(x_n) \end{bmatrix}.$$

*Then, it has the multiplicative inverse $\mathbf{S}^{-1} = \frac{1}{n}\mathrm{diag}\{1, 2, 2, \ldots, 2\}\mathbf{S}^{\top}$.*

Next, we construct the first order integration matrix by letting $j$ and $M$ be natural numbers, $u_i(x)$ be an approximate solution of $y_i(x)$ contained in (1) and (2) which is defined by a linear combination of the shifted Chebyshev polynomials $S_0(x)$, $S_1(x)$, $S_2(x)$, $\ldots$, $S_{M-1}(x)$. Then, we have

$$u_i(x) = \sum_{n=0}^{M-1} c_{n_i} S_n(x), \tag{8}$$

where $c_{n_i}$ is the unknown coefficient to be considered. Let $x_k$ be grid points generated by the zeros of the shifted Chebyshev polynomial $S_M$ defined by (5) in ascending order. When we substitute each $x_k$ into (8), they can be expressed in the matrix form

$$\begin{bmatrix} u_i(x_1) \\ u_i(x_2) \\ \vdots \\ u_i(x_M) \end{bmatrix} = \begin{bmatrix} S_0(x_1) & S_1(x_1) & \cdots & S_{M-1}(x_1) \\ S_0(x_2) & S_1(x_2) & \cdots & S_{M-1}(x_2) \\ \vdots & \vdots & \ddots & \vdots \\ S_0(x_M) & S_1(x_M) & \cdots & S_{M-1}(x_M) \end{bmatrix} \begin{bmatrix} c_{0_i} \\ c_{1_i} \\ \vdots \\ c_{M-1_i} \end{bmatrix},$$

which is denoted by $\mathbf{u}_i = \mathbf{S}\mathbf{c}_i$. Since $\mathbf{S}$ is invertible by Lemma 1(iv), we have $\mathbf{c}_i = \mathbf{S}^{-1}\mathbf{u}_i$. Now, we consider the single integration of $u_i$ from 0 to $x_k$, denoted $U_i^{(1)}(x_k)$, to obtain

$$U_i^{(1)}(x_k) = \int_0^{x_k} u_i(\xi)\, d\xi = \sum_{n=0}^{M-1} c_{n_i} \int_0^{x_k} S_n(\xi)\, d\xi = \sum_{n=0}^{M-1} c_{n_i} \overline{S}_n(x_k)$$

where $\overline{S}_n$ is denoted to the single-layer integration of $S_n$ that can explicitly find by (7) depending on its degree $n$. After substituting each node $x_k$ into the above equation, it can be written in the matrix form

$$\begin{bmatrix} U_i^{(1)}(x_1) \\ U_i^{(1)}(x_2) \\ \vdots \\ U_i^{(1)}(x_M) \end{bmatrix} = \begin{bmatrix} \overline{S}_0(x_1) & \overline{S}_1(x_1) & \cdots & \overline{S}_{M-1}(x_1) \\ \overline{S}_0(x_2) & \overline{S}_1(x_2) & \cdots & \overline{S}_{M-1}(x_2) \\ \vdots & \vdots & \ddots & \vdots \\ \overline{S}_0(x_M) & \overline{S}_1(x_M) & \cdots & \overline{S}_{M-1}(x_M) \end{bmatrix} \begin{bmatrix} c_{0_i} \\ c_{1_i} \\ \vdots \\ c_{M-1_i} \end{bmatrix},$$

which is denoted by $\mathbf{U}_i^{(1)} = \overline{\mathbf{S}}\mathbf{c}_i = \overline{\mathbf{S}}\mathbf{S}^{-1}\mathbf{u}_i := \mathbf{A}\mathbf{u}_i$, where $\mathbf{A} = \overline{\mathbf{S}}\mathbf{S}^{-1} := [a_{kl}]_{M\times M}$ is the integral operational matrix called the first order shifted Chebyshev integration matrix (SCIM). It can be also expressed to another form

$$U_i^{(1)}(x_k) = \int_0^{x_k} u_i(\xi)\, d\xi = \sum_{l=1}^{M} a_{kl} u_i(x_l). \tag{9}$$

**Remark 1** ([20]). *Based on (9), the m-layer integration of $u_i$ from 0 to $x_k$, denoted $U_i^{(m)}(x_k)$, can be easily obtained by the first order SCIM $\mathbf{A}$ multiplied by itself m times, i.e.,*

$$U_i^{(m)}(x_k) = \int_0^{x_k} \int_0^{\xi_m} \cdots \int_0^{\xi_3} \int_0^{\xi_2} u_i(\xi_1)\, d\xi_1 d\xi_2 \ldots \xi_{m-1}\xi_m \longmapsto \mathbf{U}_i^{(m)} = \mathbf{A}^m \mathbf{u}_i.$$

### 3. Procedure for Solving System of FIDEs

In this section, we devise the numerical procedure for solving the system of FIDEs (1) and verify accurate solutions obtained from this proposed procedure by comparing with other existing methods and their analytical solutions. Before constructing the procedure, we attempt to create an operational matrix of integrals for the fractional integration and derivative. Let us provide some basic definitions and properties of fractional calculus theory from [25,28] as follows.

**Definition 2.** *Let p, μ and x be real numbers such that $x > 0$ and*

$$C_\mu = \{u : (0,\infty) \to \mathbb{R} \mid u(x) = x^p u_1(x), \text{ where } u_1 \in C[0,\infty) \text{ and } p > \mu\}.$$

*The Riemann-Liouville fractional integral operator of order α of $u \in C_\mu$, $\mu \geq -1$ is defined by*

$$I^\alpha u(x) = \begin{cases} \frac{1}{\Gamma(\alpha)} \int_0^x \frac{u(t)}{(x-t)^{1-\alpha}} dt & \text{for } \alpha > 0, \\ u(x) & \text{for } \alpha = 0, \end{cases}$$

*where $\Gamma(\cdot)$ is the well-known Gamma function.*

**Definition 3.** *The Caputo fractional derivative $D^\alpha$ of $u \in C_{-1}^n$ for $n \in \mathbb{N}$ is defined by*

$$D^\alpha u(x) = I^{n-\alpha} D^n u(x) = \begin{cases} \frac{1}{\Gamma(n-\alpha)} \int_0^x \frac{u^{(n)}(t)}{(x-t)^{1-n+\alpha}} dt & \text{for } \alpha \in (n-1, n), \\ u^{(n)}(x) & \text{for } \alpha = n. \end{cases}$$

Actually, we have known that the Riemann-Liouville fractional integral operator $I^\alpha$ is a linear operator which has some important properties including

$$I^\alpha D^\beta u(x) = I^{\alpha-\beta} u(x) \text{ for } \alpha \geq \beta \quad \text{and} \tag{10}$$

$$I^\alpha x^k = \frac{\Gamma(k+1)}{\Gamma(k+\alpha+1)} x^{k+\alpha} \text{ for } k \in \mathbb{N} \cup \{0\}. \tag{11}$$

Recall that for $\alpha \in \mathbb{N}$, the Caputo differential operator $D^\alpha$ coincides with the usual differential operator of integer order. More properties can be found in [25,28].

#### 3.1. Operational Matrix of Fractional Integration

In this work, the fractional derivatives of the system of FIDEs (1) are studied based on the Caputo sense stated in Definition 3. We create the the operational matrix of fractional integrals by using the shifted Chebyshev expansion which is called the shifted Chebyshev fractional matrix (SCFM). First, we express and prove the analytic formula of the Riemann-Liouville fractional integral with order α for the shifted Chebyshev polynomials as demonstrated in Theorem 1.

**Theorem 1.** *Let $S_n(x)$ be a shifted Chebyshev polynomial. Then,*

$$I^\alpha S_n(x) = \begin{cases} \frac{x^\alpha}{\Gamma(\alpha+1)} & \text{for } n = 0, \\ n \sum\limits_{k=0}^{n} \frac{(-1)^{n-k}(n+k-1)!\,\Gamma(k+1)\,4^k}{(n-k)!(2k)!\,\Gamma(k+\alpha+1)\,L^k} x^{k+\alpha} & \text{for } n > 0. \end{cases} \tag{12}$$

**Proof of Theorem 1.** For $n = 0$, it is obvious that $S_0(x) = 1$. Then, using Definition 2 for $\alpha > 0$, we get

$$I^\alpha S_0(x) = \frac{1}{\Gamma(\alpha)} \int_0^x (x-t)^{\alpha-1} dt = \frac{1}{\Gamma(\alpha)} \left( \frac{x^\alpha}{\alpha} \right) = \frac{x^\alpha}{\Gamma(\alpha+1)}. \tag{13}$$

In addition, for $n > 0$, we have known that the fractional integration in the Riemann-Liouville sense is a linear operation. Thus, by employing (4) and (11), we obtain

$$
\begin{aligned}
I^\alpha S_n(x) &= n \sum_{k=0}^{n} \frac{(-1)^{n-k}(n+k-1)!\, 2^{2k}}{(n-k)!(2k)!\, L^k} I^\alpha(x^k) \\
&= n \sum_{k=0}^{n} \frac{(-1)^{n-k}(n+k-1)!\, 2^{2k}}{(n-k)!(2k)!\, L^k} \cdot \frac{\Gamma(k+1)}{\Gamma(k+\alpha+1)} x^{k+\alpha} \\
&= n \sum_{k=0}^{n} \frac{(-1)^{n-k}(n+k-1)!\, \Gamma(k+1)\, 4^k}{(n-k)!(2k)!\, \Gamma(k+\alpha+1)\, L^k} x^{k+\alpha}.
\end{aligned}
\tag{14}
$$

Hence, a combination of (13) and (14) leads to the desired result (12). $\quad\square$

Next, we construct an operational matrix of the Riemann-Liouville fractional integral with order $\alpha$ for an approximate unknown function $u_i(x)$ by utilizing the shifted Chebyshev expansion (8) which is called the SCFM as expressed in Theorems 2 and 3.

**Theorem 2.** *Let $u_i(x)$ be approximated by the shifted Chebyshev expansion (8). Then,*

$$
I^\alpha u_i(x) := \mathbf{J}^\alpha(x)\mathbf{S}^{-1}\mathbf{u}_i,
\tag{15}
$$

*where $\mathbf{J}^\alpha(x) := [I^\alpha S_0(x), I^\alpha S_1(x), I^\alpha S_2(x), \ldots, I^\alpha S_{M-1}(x)]$ is an M row vector in which each component can be calculated by (12), $\mathbf{u}_i = [u_i(x_1), u_i(x_2), u_i(x_3), \ldots, u_i(x_M)]^\top$ is an M column vector of each $u_i(x)$ at the zeros $x_k$ in (5) and $\mathbf{S}^{-1}$ is defined in Lemma 1(iv).*

**Proof of Theorem 2.** Since the Riemann-Liouville's fractional integration of order $\alpha > 0$ is a linear operation and use the linear combination $u_i(x)$ in (8), we have

$$
I^\alpha u_i(x) = \frac{1}{\Gamma(\alpha)} \int_0^x \frac{\sum_{n=0}^{M-1} c_{n_i} S_n(t)}{(x-t)^{1-\alpha}} dt = \sum_{n=0}^{M-1} \frac{c_{n_i}}{\Gamma(\alpha)} \int_0^x \frac{S_n(t)}{(x-t)^{1-\alpha}} dt = \sum_{n=0}^{M-1} c_{n_i} I^\alpha S_n(x).
$$

Rewriting the above equation as a vector form and use the vector $\mathbf{c}_i$ in Section 2, we get

$$
I^\alpha u_i(x) = [I^\alpha S_0(x), I^\alpha S_1(x), \ldots, I^\alpha S_{M-1}(x)][c_{0_i}, c_{1_i}, \ldots, c_{M-1_i}]^\top := \mathbf{J}^\alpha(x)\mathbf{c}_i = \mathbf{J}^\alpha(x)\mathbf{S}^{-1}\mathbf{u}_i.
$$

Hence, we obtain the desired representation (15). $\quad\square$

**Theorem 3.** *Let $\mathbf{u}_i$ be an M column vector of each $u_i(x)$ at the zeros $x_k$ in (5). Then,*

$$
I^\alpha \mathbf{u}_i := \mathbf{J}^\alpha \mathbf{S}^{-1}\mathbf{u}_i,
$$

*where $\mathbf{J}^\alpha \mathbf{S}^{-1}$ is the $M \times M$ SCFM of order $\alpha$ representing the operator $I^\alpha$, $\mathbf{S}^{-1}$ is defined in Lemma 1(iv), $\mathbf{u}_i = [u_i(x_1), u_i(x_2), u_i(x_3), \ldots, u_i(x_M)]^\top$ and*

$$
\mathbf{J}^\alpha = \begin{bmatrix} \mathbf{J}^\alpha(x_1) \\ \mathbf{J}^\alpha(x_2) \\ \vdots \\ \mathbf{J}^\alpha(x_M) \end{bmatrix} = \begin{bmatrix} I^\alpha S_0(x_1) & I^\alpha S_1(x_1) & \cdots & I^\alpha S_{M-1}(x_1) \\ I^\alpha S_0(x_2) & I^\alpha S_1(x_2) & \cdots & I^\alpha S_{M-1}(x_2) \\ \vdots & \vdots & \ddots & \vdots \\ I^\alpha S_0(x_M) & I^\alpha S_1(x_M) & \cdots & I^\alpha S_{M-1}(x_M) \end{bmatrix}.
$$

**Proof of Theorem 3.** Let $\alpha > 0$ and $M \in \mathbf{N}$. By employing the relation (15), we obtain

$$
I^\alpha \mathbf{u}_i = I^\alpha \begin{bmatrix} u_i(x_1) \\ u_i(x_2) \\ \vdots \\ u_i(x_M) \end{bmatrix} = \begin{bmatrix} I^\alpha u_i(x_1) \\ I^\alpha u_i(x_2) \\ \vdots \\ I^\alpha u_i(x_M) \end{bmatrix} = \begin{bmatrix} \mathbf{J}^\alpha(x_1)\mathbf{S}^{-1}\mathbf{u}_i \\ \mathbf{J}^\alpha(x_2)\mathbf{S}^{-1}\mathbf{u}_i \\ \vdots \\ \mathbf{J}^\alpha(x_M)\mathbf{S}^{-1}\mathbf{u}_i \end{bmatrix} = \begin{bmatrix} \mathbf{J}^\alpha(x_1) \\ \mathbf{J}^\alpha(x_2) \\ \vdots \\ \mathbf{J}^\alpha(x_M) \end{bmatrix} \mathbf{S}^{-1}\mathbf{u}_i = \mathbf{J}^\alpha \mathbf{S}^{-1}\mathbf{u}_i.
$$

Thus, we achieve the operational matrix of Riemann-Liouville fractional integral. Note that, the elements contained in $\mathbf{J}^\alpha$ can be computed by (12).   □

However, we observe that for the order $\alpha = 1$, we have $\mathbf{J}^\alpha = \overline{\mathbf{S}}$ which is defined in Section 2. Thus, when $\alpha$ is a positive integer, then $\mathbf{A}^\alpha = \mathbf{J}^\alpha \mathbf{S}^{-1}$. In order to reduce the computational time for the positive integer order $\alpha$, we consume the SCIM $\mathbf{A}^\alpha$ instead of the SCFM $\mathbf{J}^\alpha \mathbf{S}^{-1}$. Because, when the order $\alpha$ has changed, the matrix $\mathbf{J}^\alpha$ needs to recalculate its elements again. Conversely, for the matrix $\mathbf{A}^\alpha$, the SCIM $\mathbf{A}$ is computed only once, then it is raised to the power $\alpha$. Hence, we obtain the following Remark 2.

**Remark 2.** *If $\alpha \in \mathbb{N}$, $I^\alpha \mathbf{u}_i = \mathbf{A}^\alpha \mathbf{u}_i$, where $\mathbf{A} = \overline{\mathbf{S}}\mathbf{S}^{-1}$ is the SCIM defined in Section 2.*

*3.2. Numerical Solution for System of FIDEs*

The objective of this section is to create a numerical procedure for solving the system of FIDEs (1) with the given initial conditions. First, let $u_i(x)$ be an approximate solution of $y_i(x)$ for (1). Thus, (1) can be written in the form

$$D^{\alpha_i} u_i(x) = f_i(x) + \sum_{j=1}^m \left( p_{ij}(x) u_j(x) + \int_0^x \kappa_{ij}(x,t) u_j(t) dt \right), \quad x \in [0, L] \tag{16}$$

for all $i \in \{1, 2, 3, \ldots, m\}$ and $L \in \mathbb{R}^+$ with the initial conditions

$$u_i^{(v)}(0) = b_{v_i} \in \mathbb{R}, \quad v \in \{0, 1, 2, \ldots, \lceil \alpha_i \rceil - 1\}. \tag{17}$$

First, we discretize the domain $[0, L]$ into $M$ grid nodes which are generated by the zeros of $S_M$ defined in (5), where $x_1 < x_2 < x_3 < \cdots < x_M$. Then, the approximate solution $u_i(x)$ is sought at these zeros. To simplify (16), we denote each integral term by

$$Q_{ij}(x) := \int_0^x \kappa_{ij}(x,t) u_j(t) \, dt \tag{18}$$

for $j \in \{1, 2, 3, \ldots, m\}$. Thus, (16) becomes

$$D^{\alpha_i} u_i(x) = f_i(x) + \sum_{j=1}^m \left( p_{ij}(x) u_j(x) + Q_{ij}(x) \right). \tag{19}$$

Next, we attempt to eliminate the fractional derivatives from (19) by taking the $\lceil \alpha_i \rceil$-layer integrals from 0 to a zero $x_k$ on both sides of (19). Then, we have

$$\int_0^{x_k} \int_0^{\xi_{\lceil \alpha_i \rceil}} \cdots \int_0^{\xi_3} \int_0^{\xi_2} D^{\alpha_i} u_i(\xi_1) \, d\xi_1 d\xi_2 \ldots d\xi_{\lceil \alpha_i \rceil - 1} d\xi_{\lceil \alpha_i \rceil} + \sum_{l=1}^{\lceil \alpha_i \rceil} \frac{d_{il} x_k^{\lceil \alpha_i \rceil - l}}{(\lceil \alpha_i \rceil - l)!}$$

$$= \int_0^{x_k} \int_0^{\xi_{\lceil \alpha_i \rceil}} \cdots \int_0^{\xi_3} \int_0^{\xi_2} \left( f_i(\xi_1) + \sum_{j=1}^m \left( p_{ij}(\xi_1) u_j(\xi_1) + Q_{ij}(\xi_1) \right) \right) d\xi_1 d\xi_2 \ldots d\xi_{\lceil \alpha_i \rceil - 1} d\xi_{\lceil \alpha_i \rceil},$$

where $d_{i1}, d_{i2}, d_{i3}, \ldots, d_{i\lceil \alpha_i \rceil}$ are arbitrary constants emerged in the process of integrations. Conveniently, we can rewrite the above equation in another form by using the integral operator $I$ instead. Then, it becomes

$$I^{\lceil \alpha_i \rceil - \alpha_i} u_i(x_k) + \sum_{l=1}^{\lceil \alpha_i \rceil} \frac{d_{il} x_k^{\lceil \alpha_i \rceil - l}}{(\lceil \alpha_i \rceil - l)!} = I^{\lceil \alpha_i \rceil} f_i(x_k) + \sum_{j=1}^m \left( I^{\lceil \alpha_i \rceil} p_{ij}(x_k) u_j(x_k) + I^{\lceil \alpha_i \rceil} Q_{ij}(x_k) \right), \tag{20}$$

where the integration of Caputo fractional derivative term has gotten by using (10).

After that, let us consider each term contained in (20) and reformulate it to the matrix form by varying the variable $x_k$ as the zeros $x_1, x_2, x_3, \ldots, x_M$ which uses the notation "$\longmapsto$" for representing the mapping from $\mathbb{R}$ to $\mathbb{R}^M$. Then, we have

$$
I^{\lceil \alpha_i \rceil - \alpha_i} u_i(x_k) \longmapsto \begin{bmatrix} I^{\lceil \alpha_i \rceil - \alpha_i} u_i(x_1) \\ I^{\lceil \alpha_i \rceil - \alpha_i} u_i(x_2) \\ \vdots \\ I^{\lceil \alpha_i \rceil - \alpha_i} u_i(x_M) \end{bmatrix} := I^{\lceil \alpha_i \rceil - \alpha_i} \mathbf{u}_i = \mathbf{J}^{\lceil \alpha_i \rceil - \alpha_i} \mathbf{S}^{-1} \mathbf{u}_i, \tag{21}
$$

where $\mathbf{J}^{\lceil \alpha_i \rceil - \alpha_i}$ and $\mathbf{S}^{-1}$ are defined in Theorem 3 and Lemma 1(iv), respectively. Next, we consider the summation term of the unknown constants in (20), that is,

$$
\sum_{l=1}^{\lceil \alpha_i \rceil} \frac{d_{il} x_k^{\lceil \alpha_i \rceil - l}}{(\lceil \alpha_i \rceil - l)!} \longmapsto \begin{bmatrix} \frac{x_1^{\lceil \alpha_i \rceil - 1}}{(\lceil \alpha_i \rceil - 1)!} & \frac{x_1^{\lceil \alpha_i \rceil - 2}}{(\lceil \alpha_i \rceil - 2)!} & \cdots & 1 \\ \frac{x_2^{\lceil \alpha_i \rceil - 1}}{(\lceil \alpha_i \rceil - 1)!} & \frac{x_2^{\lceil \alpha_i \rceil - 2}}{(\lceil \alpha_i \rceil - 2)!} & \cdots & 1 \\ \vdots & \vdots & \ddots & \vdots \\ \frac{x_M^{\lceil \alpha_i \rceil - 1}}{(\lceil \alpha_i \rceil - 1)!} & \frac{x_M^{\lceil \alpha_i \rceil - 2}}{(\lceil \alpha_i \rceil - 2)!} & \cdots & 1 \end{bmatrix} \begin{bmatrix} d_{i1} \\ d_{i2} \\ \vdots \\ d_{i \lceil \alpha_i \rceil} \end{bmatrix} := \mathbf{X}_i \mathbf{d}_i. \tag{22}
$$

Then, the integration of the forcing terms in (20) is formulated by using Remark 2 to the matrix form

$$
I^{\lceil \alpha_i \rceil} f_i(x_k) \longmapsto \begin{bmatrix} I^{\lceil \alpha_i \rceil} f_i(x_1) \\ I^{\lceil \alpha_i \rceil} f_i(x_2) \\ \vdots \\ I^{\lceil \alpha_i \rceil} f_i(x_M) \end{bmatrix} = I^{\lceil \alpha_i \rceil} \begin{bmatrix} f_i(x_1) \\ f_i(x_2) \\ \vdots \\ f_i(x_M) \end{bmatrix} := I^{\lceil \alpha_i \rceil} \mathbf{f}_i = \mathbf{A}^{\lceil \alpha_i \rceil} \mathbf{f}_i. \tag{23}
$$

After that, we consider the remaining terms in the summation on the right-hand-side of (20). Then, its first term is converted to the matrix form

$$
I^{\lceil \alpha_i \rceil} p_{ij}(x_k) u_j(x_k) \longmapsto \begin{bmatrix} I^{\lceil \alpha_i \rceil} p_{ij}(x_1) u_j(x_1) \\ I^{\lceil \alpha_i \rceil} p_{ij}(x_2) u_j(x_2) \\ \vdots \\ I^{\lceil \alpha_i \rceil} p_{ij}(x_M) u_j(x_M) \end{bmatrix} = I^{\lceil \alpha_i \rceil} \begin{bmatrix} p_{ij}(x_1) u_j(x_1) \\ p_{ij}(x_2) u_j(x_2) \\ \vdots \\ (x_M) u_j(x_M) \end{bmatrix} := \mathbf{A}^{\lceil \alpha_i \rceil} \mathbf{P}_{ij} \mathbf{u}_j, \quad (24)
$$

where $\mathbf{P}_{ij} = \text{diag}\{p_{ij}(x_1), p_{ij}(x_2), \ldots, p_{ij}(x_M)\}$ and $\mathbf{u}_j = [u_j(x_1), u_j(x_2), \ldots, u_j(x_M)]^\top$. For another term in the summation on the right-hand-side of (20), $I^{\lceil \alpha_i \rceil} Q_{ij}(x_k)$, before transforming it to the matrix form, we consider $Q_{ij}(x_k)$ by using (9) and (18) to obtain

$$
Q_{ij}(x_k) = \int_0^{x_k} \kappa_{ij}(x_k, t) u_j(t)\, dt = \sum_{l=1}^{M} a_{kl} \kappa_{ij}(x_k, x_l) u_j(x_l),
$$

where $a_{kl}$ is an element at the $k$th row and the $l$th column of the SCIM $\mathbf{A} = \overline{\mathbf{S}} \mathbf{S}^{-1}$ defined in Section 2. Thus, we have the column vector of $Q_{ij}(x_k)$ as

$$
\begin{bmatrix} Q_{ij}(x_1) \\ Q_{ij}(x_2) \\ \vdots \\ Q_{ij}(x_M) \end{bmatrix} = \begin{bmatrix} a_{11} \kappa_{ij}(x_1, x_1) & a_{12} \kappa_{ij}(x_1, x_2) & \cdots & a_{1M} \kappa_{ij}(x_1, x_M) \\ a_{21} \kappa_{ij}(x_2, x_1) & a_{22} \kappa_{ij}(x_2, x_2) & \cdots & a_{2M} \kappa_{ij}(x_2, x_M) \\ \vdots & \vdots & \ddots & \vdots \\ a_{M1} \kappa_{ij}(x_M, x_1) & a_{M2} \kappa_{ij}(x_M, x_2) & \cdots & a_{MM} \kappa_{ij}(x_M, x_M) \end{bmatrix} \begin{bmatrix} u_j(x_1) \\ u_j(x_2) \\ \vdots \\ u_j(x_M) \end{bmatrix}
$$

which is denoted by $\mathbf{Q}_{ij} = (\mathbf{A} \odot \mathbf{K}_{ij}) \mathbf{u}_j$, where $\mathbf{A} = \overline{\mathbf{S}} \mathbf{S}^{-1} = [a_{kl}]$ is the $M \times M$ SCIM and $\odot$ is the Hadamard product defined in [29].

From the above relation and Remark 2, we then have

$$I^{\lceil \alpha_i \rceil} Q_{ij}(x_k) \longmapsto \begin{bmatrix} I^{\lceil \alpha_i \rceil} Q_{ij}(x_1) \\ I^{\lceil \alpha_i \rceil} Q_{ij}(x_2) \\ \vdots \\ I^{\lceil \alpha_i \rceil} Q_{ij}(x_M) \end{bmatrix} = I^{\lceil \alpha_i \rceil} \begin{bmatrix} Q_{ij}(x_1) \\ Q_{ij}(x_2) \\ \vdots \\ Q_{ij}(x_M) \end{bmatrix} := \mathbf{A}^{\lceil \alpha_i \rceil}(\mathbf{A} \odot \mathbf{K}_{ij})\mathbf{u}_j. \quad (25)$$

Consequently, by substituting expressions (21)–(25) into (20), we obtain

$$\mathbf{J}^{\lceil \alpha_i \rceil - \alpha_i} \mathbf{S}^{-1} \mathbf{u}_i + \mathbf{X}_i \mathbf{d}_i = \mathbf{A}^{\lceil \alpha_i \rceil} \mathbf{f}_i + \sum_{j=1}^{m} \left( \mathbf{A}^{\lceil \alpha_i \rceil} \mathbf{P}_{ij} \mathbf{u}_j + \mathbf{A}^{\lceil \alpha_i \rceil}(\mathbf{A} \odot \mathbf{K}_{ij})\mathbf{u}_j \right)$$

or it can be simplified as

$$\mathbf{J}^{\lceil \alpha_i \rceil - \alpha_i} \mathbf{S}^{-1} \mathbf{u}_i + \mathbf{X}_i \mathbf{d}_i = \mathbf{A}^{\lceil \alpha_i \rceil} \mathbf{f}_i + \mathbf{A}^{\lceil \alpha_i \rceil} \sum_{j=1}^{m} \mathbf{H}_{ij} \mathbf{u}_j, \quad (26)$$

where $\mathbf{H}_{ij} := \mathbf{P}_{ij} + (\mathbf{A} \odot \mathbf{K}_{ij})$. Finally, we vary all indices $i \in \{1, 2, 3, \ldots, m\}$ in (26). Then, we have the following system

$$\mathbf{J}^{\lceil \alpha_1 \rceil - \alpha_1} \mathbf{S}^{-1} \mathbf{u}_1 + \mathbf{X}_1 \mathbf{d}_1 = \mathbf{A}^{\lceil \alpha_1 \rceil} \mathbf{f}_1 + \mathbf{A}^{\lceil \alpha_1 \rceil}(\mathbf{H}_{11}\mathbf{u}_1 + \mathbf{H}_{12}\mathbf{u}_2 + \cdots + \mathbf{H}_{1m}\mathbf{u}_m)$$

$$\mathbf{J}^{\lceil \alpha_2 \rceil - \alpha_2} \mathbf{S}^{-1} \mathbf{u}_2 + \mathbf{X}_2 \mathbf{d}_2 = \mathbf{A}^{\lceil \alpha_2 \rceil} \mathbf{f}_2 + \mathbf{A}^{\lceil \alpha_2 \rceil}(\mathbf{H}_{21}\mathbf{u}_1 + \mathbf{H}_{22}\mathbf{u}_2 + \cdots + \mathbf{H}_{2m}\mathbf{u}_m)$$

$$\vdots$$

$$\mathbf{J}^{\lceil \alpha_m \rceil - \alpha_m} \mathbf{S}^{-1} \mathbf{u}_m + \mathbf{X}_m \mathbf{d}_m = \mathbf{A}^{\lceil \alpha_m \rceil} \mathbf{f}_m + \mathbf{A}^{\lceil \alpha_m \rceil}(\mathbf{H}_{m1}\mathbf{u}_1 + \mathbf{H}_{m2}\mathbf{u}_2 + \cdots + \mathbf{H}_{mm}\mathbf{u}_m)$$

which the system can be rearranged to the block-matrix of the form

$$\left[ \mathbf{J}(\mathbf{I}_m \otimes \mathbf{S}^{-1}) - \mathbf{B}\mathbf{H} \right] \mathbf{u} + \mathbf{X}\mathbf{d} = \mathbf{B}\mathbf{f}, \quad (27)$$

where $\mathbf{I}_m$ is an $m \times m$ identity matrix, $\mathbf{S}^{-1}$ is defined in Lemma 1(iv) and $\otimes$ is the Kronecker product defined in [29]. Other parameters contained in (27) are defined by the following block matrices:

$$\begin{aligned}
\mathbf{J} &:= \text{blkdiag}\left\{\mathbf{J}^{\lceil \alpha_1 \rceil - \alpha_1}, \mathbf{J}^{\lceil \alpha_2 \rceil - \alpha_2}, \mathbf{J}^{\lceil \alpha_3 \rceil - \alpha_3} \ldots, \mathbf{J}^{\lceil \alpha_m \rceil - \alpha_m}\right\}_{mM \times mM\prime} \\
\mathbf{B} &:= \text{blkdiag}\left\{\mathbf{A}^{\lceil \alpha_1 \rceil}, \mathbf{A}^{\lceil \alpha_2 \rceil}, \mathbf{A}^{\lceil \alpha_3 \rceil}, \ldots, \mathbf{A}^{\lceil \alpha_m \rceil}\right\}_{mM \times mM\prime} \\
\mathbf{X} &:= \text{blkdiag}\left\{\mathbf{X}_1, \mathbf{X}_2, \mathbf{X}_3, \ldots, \mathbf{X}_m\right\}_{mM \times \sum_{i=1}^{m} \lceil \alpha_i \rceil\prime} \\
\mathbf{d} &:= \left[\mathbf{d}_1, \mathbf{d}_2, \mathbf{d}_3, \ldots, \mathbf{d}_m\right]^{\top}_{\sum_{i=1}^{m} \lceil \alpha_i \rceil \times 1\prime} \\
\mathbf{u} &:= \left[\mathbf{u}_1, \mathbf{u}_2, \mathbf{u}_3, \ldots, \mathbf{u}_m\right]^{\top}_{mM \times 1\prime} \\
\mathbf{f} &:= \left[\mathbf{f}_1, \mathbf{f}_2, \mathbf{f}_3, \ldots, \mathbf{f}_m\right]^{\top}_{mM \times 1}
\end{aligned}$$

and

$$\mathbf{H} := \begin{bmatrix} \mathbf{P}_{11} + (\mathbf{A} \odot \mathbf{K}_{11}) & \mathbf{P}_{12} + (\mathbf{A} \odot \mathbf{K}_{12}) & \cdots & \mathbf{P}_{1m} + (\mathbf{A} \odot \mathbf{K}_{1m}) \\ \mathbf{P}_{21} + (\mathbf{A} \odot \mathbf{K}_{21}) & \mathbf{P}_{22} + (\mathbf{A} \odot \mathbf{K}_{22}) & \cdots & \mathbf{P}_{2m} + (\mathbf{A} \odot \mathbf{K}_{2m}) \\ \vdots & \vdots & \ddots & \vdots \\ \mathbf{P}_{m1} + (\mathbf{A} \odot \mathbf{K}_{m1}) & \mathbf{P}_{m2} + (\mathbf{A} \odot \mathbf{K}_{m2}) & \cdots & \mathbf{P}_{mm} + (\mathbf{A} \odot \mathbf{K}_{mm}) \end{bmatrix},$$

where "blkdiag$\{\cdot\}$" is a block diagonal matrix in which the off-diagonal elements are the zero matrices. However, we can see that (27) has unknown vectors apart from $\mathbf{u}$, i.e., $\mathbf{d}$ which is emerged from the process of integration for a total of $\sum_{i=1}^{m} \lceil \alpha_i \rceil$ elements.

Therefore, we require $\sum_{i=1}^{m} \lceil \alpha_i \rceil$ equations more which are constructed by using the given initial conditions (17). At specified index $i$, we use (8) to transform these conditions

into the matrix form, we have $u_i^{(v)}(0) = \sum_{n=0}^{M-1} c_{n_i} S_n^{(v)}(0) := \mathbf{s}^{(v)}\mathbf{c}_i = \mathbf{s}^{(v)}\mathbf{S}^{-1}\mathbf{u}_i = b_{v_i}$, where $\mathbf{s}^{(v)} := [S_0^{(v)}(0), S_1^{(v)}(0), S_2^{(v)}(0), \ldots, S_{M-1}^{(v)}(0)]$ is the $M$ row vector in which its elements can be found by using (6), $\mathbf{u}_i = [u_i(x_1), u_i(x_2), u_i(x_3), \ldots, u_i(x_M)]^\top$ and $\mathbf{S}^{-1}$ is defined in Lemma 1(iv). Thus, when all derivative orders $v \in \{0, 1, 2, \ldots, \lceil \alpha_i \rceil - 1\}$ are varied in the above equation, we obtain

$$\mathbf{S}_i'\mathbf{S}^{-1}\mathbf{u}_i = \mathbf{b}_i, \tag{28}$$

where $\mathbf{b}_i := [b_{0_i}, b_{1_i}, b_{2_i}, \ldots, b_{\lceil \alpha_i \rceil - 1_i}]^\top$ and

$$\mathbf{S}_i' = \begin{bmatrix} \mathbf{s}^{(0)} \\ \mathbf{s}^{(1)} \\ \vdots \\ \mathbf{s}^{(\lceil \alpha_i \rceil - 1)} \end{bmatrix} = \begin{bmatrix} S_0(0) & S_1(0) & \cdots & S_{M-1}(0) \\ S_0'(0) & S_1'(0) & \cdots & S_{M-1}'(0) \\ \vdots & \vdots & \ddots & \vdots \\ S_0^{(\lceil \alpha_i \rceil - 1)}(0) & S_1^{(\lceil \alpha_i \rceil - 1)}(0) & \cdots & S_{M-1}^{(\lceil \alpha_i \rceil - 1)}(0) \end{bmatrix}.$$

Then, we substitute all indices $i \in \{1, 2, 3, \ldots, m\}$ in (28) and write them into the block-matrix form

$$\mathbf{S}'(\mathbf{I}_m \otimes \mathbf{S}^{-1})\mathbf{u} = \mathbf{b}, \tag{29}$$

where $\mathbf{I}_m$ is an $m \times m$ identity matrix, $\mathbf{S}' := \mathrm{blkdiag}\{\mathbf{S}_1', \mathbf{S}_2', \mathbf{S}_3', \ldots, \mathbf{S}_m'\}$ and $\mathbf{b} = [\mathbf{b}_1, \mathbf{b}_2, \mathbf{b}_3, \ldots, \mathbf{b}_m]^\top$. Since (28) has $\lceil \alpha_i \rceil$ equations, it now implies that (29) has $\sum_{i=1}^m \lceil \alpha_i \rceil$ equations. Therefore, we achieve more equations as required which resulted in the numbers of unknown variables and the numbers of equations in (27) and (29) to be precisely equivalent. Finally, we can combine both (27) and (29) in order to construct the linear system in the block-matrix form as the following

$$\begin{bmatrix} \mathbf{J}(\mathbf{I}_m \otimes \mathbf{S}^{-1}) - \mathbf{BH} & \mathbf{X} \\ \mathbf{S}'(\mathbf{I}_m \otimes \mathbf{S}^{-1}) & \mathbf{0} \end{bmatrix} \begin{bmatrix} \mathbf{u} \\ \mathbf{d} \end{bmatrix} = \begin{bmatrix} \mathbf{Bf} \\ \mathbf{b} \end{bmatrix}, \tag{30}$$

where $\mathbf{0}$ is the $\sum_{i=1}^m \lceil \alpha_i \rceil \times \sum_{i=1}^m \lceil \alpha_i \rceil$ zero matrix and each parameter contained in (30) is defined as mentioned above. Note that the amount of calculation for the linear system (30) totally consists of $mM + \sum_{i=1}^m \lceil \alpha_i \rceil$ equations which can be certainly solved by using the backslash or inverse command in MatLab solver. Now, we can seek the approximate solution $\mathbf{u}$ by solving (30). The obtained solution $\mathbf{u}$ consists of each result $\mathbf{u}_i$ for $i \in \{1, 2, 3, \ldots, m\}$ which is at the position followed the zeros of shifted Chebyshev polynomial $S_M$. Nevertheless, if we would like to know the solution $u_i(x)$ at a different position within the domain $[0, L]$, it can be calculated by

$$u_i(x) = \sum_{n=0}^{M-1} c_{n_i} S_n(x) := \mathbf{S}(x)\mathbf{c}_i = \mathbf{S}(x)\mathbf{S}^{-1}\mathbf{u}_i, \tag{31}$$

where $\mathbf{S}(x) := [S_0(x), S_1(x), S_2(x), \ldots, S_{M-1}(x)]$ and $\mathbf{u}_i$ obtains from solving (30).

### 3.3. Experimental Examples for System of FIDEs

In order to illustrate the effectiveness of the proposed numerical procedure in the preceding section, we now present some experimental examples for solving the system of FIDEs (1). In calculation, we implement the proposed method to solve four examples and show its accuracy and efficiency which is measured by the absolute error $Eu_i(x) := |u_i^*(x) - u_i(x)|$, where $u_i^*$ and $u_i$ are the analytical and approximate solutions at each $x$ in the domain, respectively, for all $i \in \{1, 2, 3, \ldots, m\}$. These examples can be used as a basis to demonstrate the applicability of the presented numerical procedure. All the experiments are carried out by MatLab R2016a on a computer equipped with a CPU Intel(R) Core(TM) i7-6700 at 3.40 GHz running on Windows 10.

**Example 1** ([24]). *Consider the following linear system of FIDEs over $x \in [0,1]$*

$$D^{\alpha_1} u_1(x) = 1 + x + x^2 - u_2(x) - \int_0^x (u_1(t) + u_2(t)) dt,$$

$$D^{\alpha_2} u_2(x) = -1 - x + u_1(x) - \int_0^x (u_1(t) - u_2(t)) dt,$$

*for $\alpha_1, \alpha_2 \in (0,1]$ with the initial conditions $u_1(0) = 1$ and $u_2(0) = -1$. The exact solutions for $\alpha_1 = \alpha_2 = 1$ are $u_1^*(x) = x + e^x$ and $u_2^*(x) = x - e^x$.*

In this Example 1, the fractional orders of derivative are considered on $(0,1]$. By employing our numerical procedure, we have the approximate solutions $u_1(x)$ and $u_2(x)$ for any $x \in [0,1]$. In Table 1, the absolute errors between the exact and approximate solutions at $\alpha_1 = \alpha_2 = 1$ are demonstrated. It also shows a comparison of the absolute errors with $M = 16$ between our suggested procedure and the technique based on operational matrices of triangular functions (OMTF) given by [24]. We can see that our method provides much higher accuracy than the OMTF. The consuming time for $M = 16$ via MatLab program is about 0.0805 s. Additionally, Figure 1a,b depict the comparisons between analytical and numerical solutions $u_1(x)$ and $u_2(x)$ with $M = 30$ and $\alpha_1 = \alpha_2 = 1$. We can see that our obtained solutions quite match exactly.

**Table 1.** Absolute errors of $u_1(x)$ and $u_2(x)$ when $\alpha_1 = \alpha_2 = 1$ for Example 1.

| | OMTF [24] | | FIM-SCP | |
| --- | --- | --- | --- | --- |
| $x$ | $Eu_1(x)$ | $Eu_2(x)$ | $Eu_1(x)$ | $Eu_2(x)$ |
| 0.1 | $5.5 \times 10^{-4}$ | $5.5 \times 10^{-4}$ | $6.4393 \times 10^{-15}$ | $1.9984 \times 10^{-15}$ |
| 0.5 | $2.7 \times 10^{-4}$ | $2.7 \times 10^{-4}$ | $1.7764 \times 10^{-15}$ | $2.6645 \times 10^{-15}$ |
| 0.9 | $1.9 \times 10^{-3}$ | $1.9 \times 10^{-3}$ | $1.7764 \times 10^{-15}$ | $1.7764 \times 10^{-15}$ |

In Table 2, absolute errors of the approximate solutions $u_1(x)$ and $u_2(x)$ are shown at different orders $\alpha_1 = \alpha_2 := \alpha \in \{0.99, 0.999, 0.9999\}$ and $M = 16$. This investigation shows that as the fractional order $\alpha$ increases from 0.99 to 0.9999, the respective accuracy is increasing and attained its maximum accuracy at $\alpha = 1$. Finally, the error analysis again verifies that at several values of $\alpha \in \{0.91, 0.93, 0.95, 0.97, 0.99, 1\}$, the obtained numerical solutions converge to the integer order solutions which their behaviors are also displayed in Figure 1c,d. They indeed attain to the blue solid line ($\alpha = 1$).

**Example 2** ([30]). *Consider the following linear system of FIDEs over $x \in [0,1]$*

$$D^{\alpha_1} u_1(x) = -1 - x^2 - \sin x + \int_0^x (u_1(t) + u_2(t)) dt,$$

$$D^{\alpha_2} u_2(x) = 1 - 2 \sin x - \cos x + \int_0^x (u_1(t) - u_2(t)) dt,$$

*for $\alpha_1, \alpha_2 \in (0,2]$ with the initial conditions $u_1(0) = 1$, $u_2(0) = 0$, $u_1'(0) = 1$ and $u_2'(0) = 2$. The analytical solutions when $\alpha_1 = \alpha_2 = 2$ are $u_1^*(x) = x + \cos x$ and $u_2^*(x) = x + \sin x$.*

**Table 2.** Absolute errors of $u_1(x)$ and $u_2(x)$ at different orders $\alpha$ for Example 1.

| | $\alpha = 0.99$ | | $\alpha = 0.999$ | | $\alpha = 0.9999$ | |
| --- | --- | --- | --- | --- | --- | --- |
| $x$ | $Eu_1(x)$ | $Eu_2(x)$ | $Eu_1(x)$ | $Eu_2(x)$ | $Eu_1(x)$ | $Eu_2(x)$ |
| 0.1 | $3.8691 \times 10^{-2}$ | $3.8203 \times 10^{-2}$ | $3.9405 \times 10^{-3}$ | $3.8926 \times 10^{-3}$ | $3.9477 \times 10^{-4}$ | $3.8999 \times 10^{-4}$ |
| 0.5 | $6.2202 \times 10^{-2}$ | $4.0183 \times 10^{-2}$ | $6.3909 \times 10^{-3}$ | $4.1382 \times 10^{-3}$ | $6.4082 \times 10^{-4}$ | $4.1504 \times 10^{-4}$ |
| 0.9 | $7.7116 \times 10^{-2}$ | $3.9661 \times 10^{-2}$ | $7.9564 \times 10^{-3}$ | $4.0861 \times 10^{-3}$ | $7.9813 \times 10^{-4}$ | $4.0984 \times 10^{-4}$ |

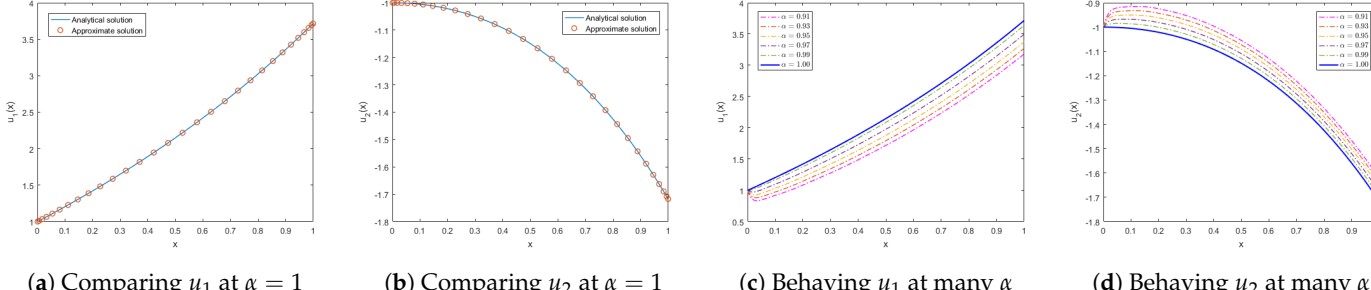

(**a**) Comparing $u_1$ at $\alpha = 1$    (**b**) Comparing $u_2$ at $\alpha = 1$    (**c**) Behaving $u_1$ at many $\alpha$    (**d**) Behaving $u_2$ at many $\alpha$

**Figure 1.** The behaviors of the approximate solutions in Example 1. (**a**,**b**) depict the comparisons between analytical and numerical solutions $u_1(x)$ and $u_2(x)$ with $M = 30$ and $\alpha_1 = \alpha_2 = 1$. The error analysis again verifies that at several values of $\alpha \in \{0.91, 0.93, 0.95, 0.97, 0.99, 1\}$, the obtained numerical solutions converge to the integer order solutions which their behaviors are also displayed in (**c**,**d**).

This experimental example is considering the problem under the fractional orders of derivatives within the interval $(0, 2]$. By using our proposed numerical procedure, we have the numerical solutions $u_1(x)$ and $u_2(x)$ for each $x \in [0, 1]$ which are compared with solutions given by the OMTF [24] measured by the absolute error as shown in Table 3. We can see that our FIM-SCP produces higher accuracy than the OMTF with the same number of discretization nodes $M = 16$ when the fractional orders $\alpha_1 = \alpha_2 = 2$. Moreover, we plot the obtained approximate solutions $u_1(x)$ and $u_2(x)$ compared to their analytical solutions as depicted in Figure 2a,b, respectively. These figures show that these solutions perfectly match with the exact values. The computational time for $M = 16$ of this process is around 0.0984 s.

**Table 3.** Absolute errors of $u_1(x)$ and $u_2(x)$ when $\alpha_1 = \alpha_2 = 2$ for Example 2.

| $x$ | OMTF [24] | | FIM-SCP | |
|---|---|---|---|---|
| | $Eu_1(x)$ | $Eu_2(x)$ | $Eu_1(x)$ | $Eu_2(x)$ |
| 0.1 | $4.6 \times 10^{-4}$ | $4.5 \times 10^{-5}$ | $1.1564 \times 10^{-12}$ | $1.0630 \times 10^{-14}$ |
| 0.5 | $3.2 \times 10^{-5}$ | $2.6 \times 10^{-4}$ | $6.2355 \times 10^{-12}$ | $8.7264 \times 10^{-14}$ |
| 0.9 | $2.2 \times 10^{-3}$ | $5.2 \times 10^{-3}$ | $1.1497 \times 10^{-11}$ | $4.2233 \times 10^{-13}$ |

Furthermore, Table 4 demonstrates the absolute errors of approximate solutions $u_1$ and $u_2$ at different values of the fractional orders $\alpha_1 = \alpha_2 := \alpha \in \{1.99, 1.999, 1.9999\}$ with $M = 16$. We can see that their errors are decreasing when $\alpha \to 2$. Finally, Figure 2c,d show the behavior of the obtained approximate solutions for the proposed system of FIDEs with the nodal point $M = 30$ for different values of the fractional orders $\alpha \in \{1.91, 1.93, 1.95, 1.97, 1.99, 2\}$. We can see that they also tend to the blue solid line, that is, the integer order solution at $\alpha = 2$.

**Table 4.** Absolute errors of $u_1(x)$ and $u_2(x)$ at different orders $\alpha$ for Example 2.

| $x$ | $\alpha = 1.99$ | | $\alpha = 1.999$ | | $\alpha = 1.9999$ | |
|---|---|---|---|---|---|---|
| | $Eu_1(x)$ | $Eu_2(x)$ | $Eu_1(x)$ | $Eu_2(x)$ | $Eu_1(x)$ | $Eu_2(x)$ |
| 0.1 | $1.6774 \times 10^{-1}$ | $4.7755 \times 10^{-3}$ | $1.7219 \times 10^{-2}$ | $4.8221 \times 10^{-4}$ | $1.7264 \times 10^{-3}$ | $4.8267 \times 10^{-5}$ |
| 0.5 | $9.6692 \times 10^{-1}$ | $3.4738 \times 10^{-2}$ | $1.0059 \times 10^{-1}$ | $3.5360 \times 10^{-3}$ | $1.0099 \times 10^{-2}$ | $3.5423 \times 10^{-4}$ |
| 0.9 | $1.8061 \times 10^{-0}$ | $2.8713 \times 10^{-2}$ | $1.8876 \times 10^{-1}$ | $2.9213 \times 10^{-3}$ | $1.8960 \times 10^{-2}$ | $2.9265 \times 10^{-4}$ |

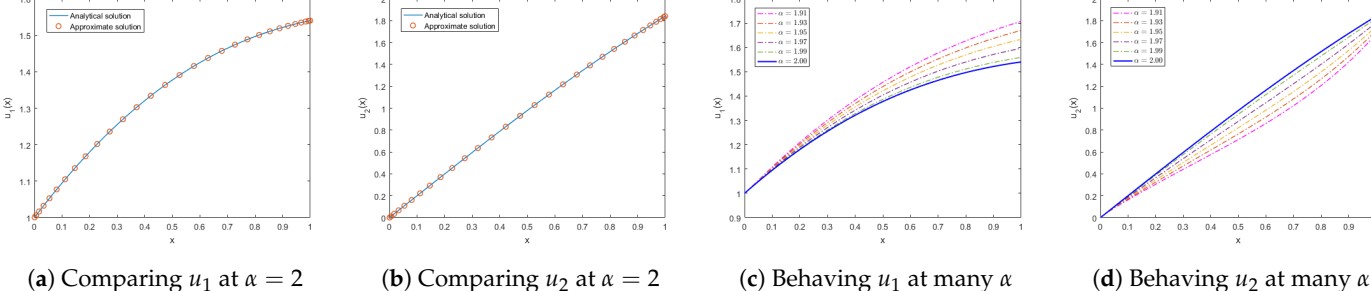

(**a**) Comparing $u_1$ at $\alpha = 2$  (**b**) Comparing $u_2$ at $\alpha = 2$  (**c**) Behaving $u_1$ at many $\alpha$  (**d**) Behaving $u_2$ at many $\alpha$

**Figure 2.** The behaviors of the approximate solutions in Example 2. We plot the obtained approximate solutions $u_1(x)$ and $u_2(x)$ compared to their analytical solutions as depicted in Figure (**a**,**b**). (**c**,**d**) show the behavior of the obtained approximate solutions for the proposed system of FIDEs with the nodal point $M = 30$ for different values of the fractional orders $\alpha \in \{1.91, 1.93, 1.95, 1.97, 1.99, 2\}$.

**Example 3** ([30])**.** *Consider the following linear system of FIDEs over $x \in [0,1]$*

$$D^{\alpha_1} u_1(x) = 2 + e^x - 3e^{2x} + e^{3x} + \int_0^x (6u_2(t) - 3u_3(t))dt,$$

$$D^{\alpha_2} u_2(x) = e^x + 2e^{2x} - e^{3x} + \int_0^x (3u_3(t) - u_1(t))dt,$$

$$D^{\alpha_3} u_3(x) = -e^x + e^{2x} + 3e^{3x} + \int_0^x (u_1(t) - 2u_2(t))dt,$$

*for $\alpha_1, \alpha_2, \alpha_3 \in (0,1]$ subject to the initial conditions $u_1(0) = u_2(0) = u_3(0) = 1$. The analytical solutions when $\alpha_1 = \alpha_2 = \alpha_3 = 1$ are $u_1^*(x) = e^x$, $u_2^*(x) = e^{2x}$ and $u_3^* = e^{3x}$.*

This system consists of three equations of FIDEs and the fractional orders are in the interval $(0,1]$. Based on the presented numerical procedure, by solving the corresponding block-matrix Equation (30) with $M = 16$ and $\alpha_1 = \alpha_2 = \alpha_3 = 1$, we obtain the numerical solutions $u_1(x)$, $u_2(x)$ and $u_3(x)$. When we find accuracy of these obtained solutions via the absolute error and compare with the OMTF [24], we can see that our method gives a much higher accuracy as shown in Table 5 and the running time is 0.1216 s. In addition, we plot the comparing graphs between approximate and exact solutions as depicted in Figure 3a–c and also the behaviors of the obtained solutions when the fractional order $\alpha \to 1$ are shown in Figure 3d–f, where $\alpha_1 = \alpha_2 = \alpha_3 := \alpha \in \{0.95, 0.96, 0.97, 0.98, 0.99, 1\}$.

From Examples 1 ($\alpha_1 = \alpha_2 = 1$), 2 ($\alpha_1 = \alpha_2 = 2$) and 3 ($\alpha_1 = \alpha_2 = \alpha_3 = 1$), we can see that our proposed procedure for solving the system of FIDEs provides an excellent accuracy in terms of the absolute error when the fractional orders tend to integer orders which can be seen in the previous tables. Moreover, we can say that a sequence $(\mathbf{u}_M)$ converges to the exact solution $\mathbf{u}^*$ with order $p$ if there exists a constant $C$ such that $\|\mathbf{u}^* - \mathbf{u}_M\| < CM^{-p}$ or $\|\mathbf{u}^* - \mathbf{u}_M\| = \mathcal{O}(M^{-p})$ using the big-O notation [31]. In practice, for approximating an order of convergence $p$, we take the natural logarithmic function on both sides of the above expression. Thus, we obtain $\ln \|\mathbf{u}^* - \mathbf{u}_M\| \approx \ln C - p \ln M$. However, if we take two distinct discretizations $M_{\text{old}}$ and $M_{\text{new}}$ into the equation, we can solve them to find the estimation of convergence order by $p \approx \dfrac{\ln(\|\mathbf{u}^* - \mathbf{u}_{M_{\text{new}}}\| / \|\mathbf{u}^* - \mathbf{u}_{M_{\text{old}}}\|)}{\ln(M_{\text{old}} / M_{\text{new}})}$, where $\mathbf{u}_{M_{\text{old}}}$ and $\mathbf{u}_{M_{\text{new}}}$ denote numerical solutions obtained by using the consecutive discretization nodes $M_{\text{old}}$ and $M_{\text{new}}$ in ascending order, respectively. Hence, the orders of convergence for Examples 1–3 based on Euclidean norm are considered numerically and presented in Table 6. It is obvious that the obtained convergence orders rapidly increase when the number of nodes $M$ ever increases.

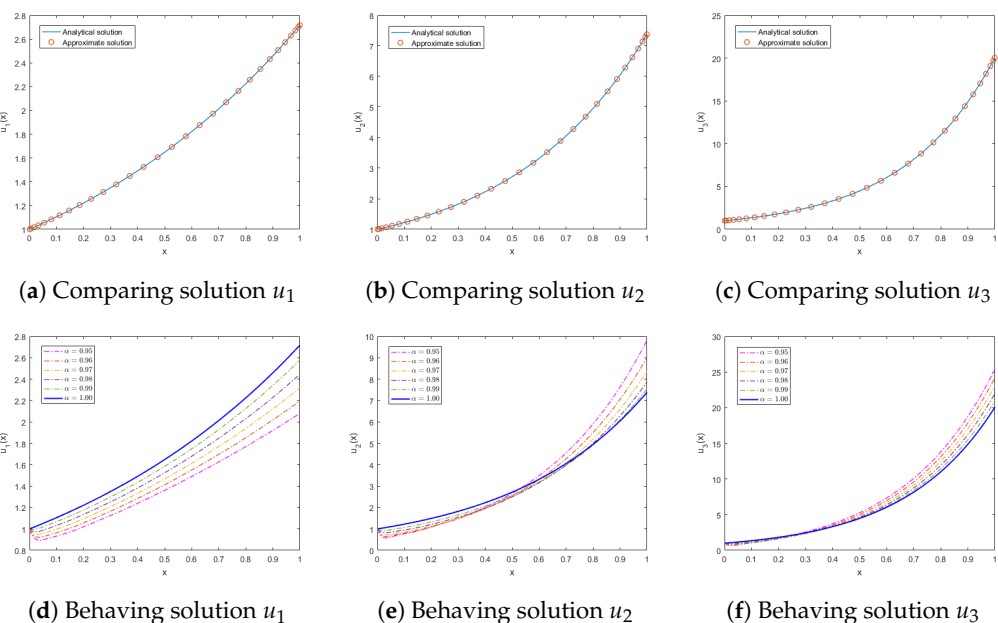

(**a**) Comparing solution $u_1$   (**b**) Comparing solution $u_2$   (**c**) Comparing solution $u_3$

(**d**) Behaving solution $u_1$   (**e**) Behaving solution $u_2$   (**f**) Behaving solution $u_3$

**Figure 3.** The behaviors of the approximate solutions in Example 3. The comparing graphs between approximate and exact solutions as depicted in (**a**–**c**) and also the behaviors of the obtained solutions when the fractional order $\alpha \to 1$ are shown in (**d**–**f**), where $\alpha_1 = \alpha_2 = \alpha_3 := \alpha \in \{0.95, 0.96, 0.97, 0.98, 0.99, 1\}$.

**Table 5.** Absolute errors of $u_1(x)$, $u_2(x)$ and $u_3(x)$ at $\alpha_1 = \alpha_2 = \alpha_3 = 1$ for Example 3.

| | OMTF [24] | | | FIM-SCP | | |
|---|---|---|---|---|---|---|
| $x$ | $Eu_1(x)$ | $Eu_2(x)$ | $Eu_3(x)$ | $Eu_1(x)$ | $Eu_2(x)$ | $Eu_3(x)$ |
| 0.1 | $5.4 \times 10^{-4}$ | $2.6 \times 10^{-3}$ | $6.6 \times 10^{-3}$ | $2.0206 \times 10^{-14}$ | $7.3275 \times 10^{-15}$ | $1.3323 \times 10^{-15}$ |
| 0.5 | $4.6 \times 10^{-4}$ | $4.9 \times 10^{-3}$ | $1.0 \times 10^{-2}$ | $1.1102 \times 10^{-14}$ | $1.7764 \times 10^{-15}$ | $2.5757 \times 10^{-14}$ |
| 0.9 | $2.6 \times 10^{-3}$ | $3.5 \times 10^{-2}$ | $9.8 \times 10^{-2}$ | $2.5313 \times 10^{-14}$ | $4.4409 \times 10^{-15}$ | $1.7764 \times 10^{-15}$ |

**Table 6.** Convergence orders of the approximate solutions for Examples 1–3.

| $M$ | Example 1 | | Example 2 | | Example 3 | |
|---|---|---|---|---|---|---|
| | $\|\mathbf{u}^* - \mathbf{u}_M\|_2$ | Order $p$ | $\|\mathbf{u}^* - \mathbf{u}_M\|_2$ | Order $p$ | $\|\mathbf{u}^* - \mathbf{u}_M\|_2$ | Order $p$ |
| 4 | $1.7668 \times 10^{-3}$ | - | $1.2217 \times 10^{-2}$ | - | $3.7762 \times 10^{-1}$ | - |
| 5 | $9.5749 \times 10^{-5}$ | 13.064 | $1.0315 \times 10^{-3}$ | 11.077 | $5.9667 \times 10^{-2}$ | 8.268 |
| 6 | $4.3033 \times 10^{-6}$ | 17.016 | $7.3615 \times 10^{-5}$ | 14.480 | $7.9496 \times 10^{-3}$ | 11.056 |
| 7 | $1.6451 \times 10^{-7}$ | 21.175 | $3.6184 \times 10^{-6}$ | 19.545 | $9.0269 \times 10^{-4}$ | 14.113 |

## 4. Procedure for Solving System of CIDEs

In this section, we extend the concept of solving system of FIDEs in Section 3 by studying in the common case, i.e., the order of the fractional derivative is focused on the positive integer which is called CIDE. Therefore, we construct the numerical procedure for solving the generalized system of CIDEs (2). An accuracy of the obtained solutions is also verified in this section.

### 4.1. Numerical Solution for System of CIDEs

Let us first introduce the system of CIDEs (2) with the given initial conditions. Let $u_j(x)$ be an approximate solution of $y_j(x)$ contained in (2). Then, it becomes

$$\sum_{j=1}^{m} \mathcal{L}_{ij} u_j(x) = f_i(x) + \sum_{j=1}^{m} \lambda_{ij} \int_0^x \kappa_{ij}(x, t) u_j(t) dt, \quad x \in [0, L] \tag{32}$$

for all $i \in \{1, 2, 3, \ldots, m\}$ and $L \in \mathbb{R}^+$ with the given initial conditions $u_i^{(v)}(0) = b_{v_i} \in \mathbb{R}$ for $v \in \{0, 1, 2, \ldots, h_i - 1\}$, where $h_i = \max_{1 \leq j \leq m} r_{ij}$ when $r_{ij}$ is the highest order of derivative for each $u_j$ contained in the linear differential operator $\mathcal{L}_{ij}$ which is defined by

$$\mathcal{L}_{ij} := p_{ij}^{\langle r_{ij} \rangle}(x) \frac{d^{r_{ij}}}{dx^{r_{ij}}} + p_{ij}^{\langle r_{ij}-1 \rangle}(x) \frac{d^{r_{ij}-1}}{dx^{r_{ij}-1}} + p_{ij}^{\langle r_{ij}-2 \rangle}(x) \frac{d^{r_{ij}-2}}{dx^{r_{ij}-2}} + \cdots + p_{ij}^{\langle 0 \rangle}(x), \tag{33}$$

where $p_{ij}^{\langle r \rangle}(x)$ for each $r \in \{0, 1, 2, \ldots, r_{ij}\}$ is continuously differentiable function up to the highest order of derivative contained in (32). We apply the idea of FIM-SCP described in Section 2 to deal with the integration term in (32). Then, the numerical procedure for solving the system of CIDEs is constructed. First, by using (18), we can rewrite (32) into

$$\sum_{j=1}^{m} \mathcal{L}_{ij} u_j(x) = f_i(x) + \sum_{j=1}^{m} \lambda_{ij} Q_{ij}(x). \tag{34}$$

Next, we discretize the domain $[0, L]$ into $M$ nodes which are generated by the zeros of $S_M$ defined in (5), i.e., $x_1 < x_2 < x_3 < \cdots < x_M$. Using the FIM-SCP, we have to eliminate all derivatives from (34). Actually, we know that the highest order of derivatives in each $i$th equation of (34) that is $h_i$. In order to remove its derivatives, the $h_i$-layer integrals from 0 to $x_k$ are taken on both sides of (34). Thus, we obtain

$$Z_{ij}(x_k) + \sum_{l=1}^{h_i} \frac{d_{il} x_k^{h_i - l}}{(h_i - l)!} = \int_0^{x_k} \int_0^{\xi_{h_i}} \cdots \int_0^{\xi_3} \int_0^{\xi_2} \left( f_i(\xi_1) + \sum_{j=1}^{m} \lambda_{ij} Q_{ij}(\xi_1) \right) d\xi_1 d\xi_2 \ldots d\xi_{h_i - 1} d\xi_{h_i}, \tag{35}$$

where $d_{i1}, d_{i2}, d_{i3}, \ldots, d_{ih_i}$ are arbitrary constants emerged in the process of integrations and $Z_{ij}(x_k)$ is an integration terms for each $u_j$. By employing (33), it can be defined by

$$Z_{ij}(x_k) := \int_0^{x_k} \int_0^{\xi_{h_i}} \cdots \int_0^{\xi_3} \int_0^{\xi_2} \left( \sum_{n=0}^{r_{ij}} p_{ij}^{\langle r_{ij}-n \rangle}(\xi_1) \frac{d^{r_{ij}-n}}{d\xi_1^{r_{ij}-n}} u_j(\xi_1) \right) d\xi_1 d\xi_2 \ldots d\xi_{h_i - 1} d\xi_{h_i},$$

where $p_{ij}^{\langle r \rangle}$ is the coefficient function corresponding to the $r$th order derivative of $u_j$ for $r \in \{0, 1, 2, \ldots, r_{ij}\}$. Next, we reformulate $Z_{ij}(x_k)$ using the integration by parts for each term to remove all derivatives from $u_j$. Anywise, $Z_{ij}(x_k)$ can be separated into two cases which are considering as $r_{ij} = h_i$ and $r_{ij} < h_i$. Thus, for the first case $r_{ij} = h_i$, it becomes

$$Z_{ij}(x_k) = \sum_{n=0}^{h_i} (-1)^n \binom{h_i}{n} \int_0^{x_k} \cdots \int_0^{\eta_2} (p_{ij}^{\langle h_i \rangle})^{(n)} u_j \, d\eta_1 \ldots d\eta_n$$

$$+ \int_0^{x_k} \left[ \sum_{n=0}^{h_i-1} (-1)^n \binom{h_i - 1}{n} \int_0^{\xi_{h_i}} \cdots \int_0^{\eta_2} (p_{ij}^{\langle h_i-1 \rangle})^{(n)} u_j \, d\eta_1 \ldots d\eta_n \right] d\xi_{h_i}$$

$$+ \int_0^{x_k} \int_0^{\xi_{h_i}} \left[ \sum_{n=0}^{h_i-2} (-1)^n \binom{h_i - 2}{n} \int_0^{\xi_{h_i-1}} \cdots \int_0^{\eta_2} (p_{ij}^{\langle h_i-2 \rangle})^{(n)} u_j \, d\eta_1 \ldots d\eta_n \right] d\xi_{h_i-1} d\xi_{h_i}$$

$$\vdots$$

$$+ \int_0^{x_k} \int_0^{\xi_{h_i}} \cdots \int_0^{\xi_3} \int_0^{\xi_2} p_{ij}^{\langle 0 \rangle} u_j \, d\xi_1 d\xi_2 \ldots d\xi_{h_i-1} d\xi_{h_i} \tag{36}$$

and for the second case $r_{ij} < h_i$, we have

$$Z_{ij}(x_k) = \int_0^{x_k} \cdots \int_0^{\xi_{r_{ij}+2}} \left[ \sum_{n=0}^{r_{ij}} (-1)^n \binom{r_{ij}}{n} \int_0^{\xi_{r_{ij}-1}} \cdots \int_0^{\eta_2} (p_{ij}^{\langle r_{ij} \rangle})^{(n)} u_j \, d\eta_1 \ldots d\eta_n \right] d\xi_{r_{ij}-1} \ldots d\xi_{h_i}$$

$$+ \int_0^{x_k} \cdots \int_0^{\xi_{r_{ij}+1}} \left[ \sum_{n=0}^{r_{ij}-1} (-1)^n \binom{r_{ij}-1}{n} \int_0^{\xi_{r_{ij}-2}} \cdots \int_0^{\eta_2} (p_{ij}^{\langle r_{ij}-1 \rangle})^{(n)} u_j \, d\eta_1 \ldots d\eta_n \right] d\xi_{r_{ij}-2} \ldots d\xi_{h_i}$$

$$+ \int_0^{x_k} \cdots \int_0^{\xi_{r_{ij}}} \left[ \sum_{n=0}^{r_{ij}-2} (-1)^n \binom{r_{ij}-2}{n} \int_0^{\xi_{r_{ij}-3}} \cdots \int_0^{\eta_2} (p_{ij}^{\langle r_{ij}-2 \rangle})^{(n)} u_j \, d\eta_1 \ldots d\eta_n \right] d\xi_{r_{ij}-3} \ldots d\xi_{h_i}$$

$$\vdots$$

$$+ \int_0^{x_k} \int_0^{\xi_{h_i}} \cdots \int_0^{\xi_3} \int_0^{\xi_2} p_{ij}^{\langle 0 \rangle} u_j \, d\xi_1 d\xi_2 \ldots d\xi_{h_i-1} d\xi_{h_i}, \tag{37}$$

where $(p_{ij}^{\langle r \rangle})^{(n)}$ is the $n$th order derivative of $p_{ij}^{\langle r \rangle}$ for $r, n \in \{0, 1, 2, \ldots, r_{ij}\}$ in both (36) and (37). Hence, (35) can be written in another form

$$\sum_{j=1}^m Z_{ij}(x_k) + \sum_{l=1}^{h_i} \frac{d_{il} x_k^{h_i-l}}{(h_i - l)!} = I^{h_i} f_i(x_k) + \sum_{j=1}^m \lambda_{ij} I^{h_i} Q_{ij}(x_k), \tag{38}$$

where $I^{h_i}$ is the $h_i$-layer repeated integral operator from 0 to the zero $x_k$. Subsequently, we apply the idea of our proposed FIM-SCP described in Section 2 to transform (38) into the matrix form by varying $x_1, x_2, x_3, \ldots, x_M$ as the zeros of shifted Chebyshev polynomial $S_M$ defined in (5). Let $Z_{ij}(x_k) \longmapsto [Z_{ij}(x_1), Z_{ij}(x_2), Z_{ij}(x_3), \ldots, Z_{ij}(x_M)]^\top := \mathbf{Z}_{ij}$. Then, for the first case $r_{ij} = h_i$, we can express (36) in the matrix form

$$\mathbf{Z}_{ij} = \left[ \sum_{n=0}^{h_i} (-1)^n \binom{h_i}{n} \mathbf{A}^n (\mathbf{P}_{ij}^{\langle h_i \rangle})^{(n)} \mathbf{u}_j \right] + \mathbf{A}^1 \left[ \sum_{n=0}^{h_i-1} (-1)^n \binom{h_i-1}{n} \mathbf{A}^n (\mathbf{P}_{ij}^{\langle h_i-1 \rangle})^{(n)} \mathbf{u}_j \right]$$

$$+ \mathbf{A}^2 \left[ \sum_{n=0}^{h_i-2} (-1)^n \binom{h_i-2}{n} \mathbf{A}^n (\mathbf{P}_{ij}^{\langle h_i-2 \rangle})^{(n)} \mathbf{u}_j \right] + \cdots + \mathbf{A}^{h_i} \mathbf{P}_{ij}^{\langle 0 \rangle} \mathbf{u}_j \tag{39}$$

and for the second case $r_{ij} < h_i$, (37) can be written in the matrix form

$$\mathbf{Z}_{ij} = \mathbf{A}^{h_i-r_{ij}} \left[ \sum_{n=0}^{r_{ij}} (-1)^n \binom{r_{ij}}{n} \mathbf{A}^n (\mathbf{P}_{ij}^{\langle r_{ij} \rangle})^{(n)} \mathbf{u}_j \right] + \mathbf{A}^{h_i-r_{ij}+1} \left[ \sum_{n=0}^{r_{ij}-1} (-1)^n \binom{r_{ij}-1}{n} \mathbf{A}^n (\mathbf{P}_{ij}^{\langle r_{ij}-1 \rangle})^{(n)} \mathbf{u}_j \right]$$

$$+ \mathbf{A}^{h_i-r_{ij}+2} \left[ \sum_{n=0}^{r_{ij}-2} (-1)^n \binom{r_{ij}-2}{n} \mathbf{A}^n (\mathbf{P}_{ij}^{\langle r_{ij}-2 \rangle})^{(n)} \mathbf{u}_j \right] + \cdots + \mathbf{A}^{h_i} \mathbf{P}_{ij}^{\langle 0 \rangle} \mathbf{u}_j, \tag{40}$$

where $(\mathbf{P}_{ij}^{\langle r \rangle})^{(n)} := \mathrm{diag}\{ (p_{ij}^{\langle r \rangle})^{(n)}(x_1), (p_{ij}^{\langle r \rangle})^{(n)}(x_2), \ldots, (p_{ij}^{\langle r \rangle})^{(n)}(x_M) \}$, $\mathbf{A} = \bar{\mathbf{S}} \mathbf{S}^{-1}$ is the SCIM described in Section 2 and $\mathbf{u}_j = [u_j(x_1), u_j(x_2), u_j(x_3), \ldots, u_j(x_M)]^\top$. However, we can explicitly see that when substituting $r_{ij} = h_i$ in (40), it indeed becomes (39). Thus, we can only use (40) which is enough to represent $\mathbf{Z}_{ij}$ for both cases, i.e., $r_{ij} \le h_i$. Moreover, (40) can be simplified to

$$\mathbf{Z}_{ij} = \sum_{k=0}^{r_{ij}} \left( \sum_{n=0}^{r_{ij}-k} (-1)^n \binom{r_{ij}-k}{n} \mathbf{A}^{n+h_i-r_{ij}+k} (\mathbf{P}_{ij}^{\langle r_{ij}-k \rangle})^{(n)} \right) \mathbf{u}_j := \mathbf{L}_{ij} \mathbf{u}_j. \tag{41}$$

Next, we transform the remaining terms of (38) into the matrix form by utilizing the same processes with (22), (23) and (25) that change $\lceil \alpha_i \rceil$ into $h_i$ instead, respectively, i.e.,

$$\sum_{l=1}^{h_i} \frac{d_{il} x_k^{h_i-l}}{(h_i - l)!} \longmapsto \mathbf{X}_i \mathbf{d}_i, \quad I^{h_i} f_i(x_k) \longmapsto \mathbf{A}^{h_i} \mathbf{f}_i, \quad I^{h_i} Q_{ij}(x_k) \longmapsto \mathbf{A}^{h_i} (\mathbf{A} \odot \mathbf{K}_{ij}) \mathbf{u}_j, \tag{42}$$

where the parameters are defined in the similar idea as presented in Section 3.2. Hence, by substituting the expressions (41) and (42) into (38), we obtain

$$\sum_{j=1}^{m} \mathbf{L}_{ij}\mathbf{u}_j + \mathbf{X}_i\mathbf{d}_i = \mathbf{A}^{h_i}\mathbf{f}_i + \sum_{j=1}^{m} \lambda_{ij}\mathbf{A}^{h_i}(\mathbf{A} \odot \mathbf{K}_{ij})\mathbf{u}_j$$

or it can be simplified to $\sum_{j=1}^{m} \mathbf{H}_{ij}\mathbf{u}_j + \mathbf{X}_i\mathbf{d}_i = \mathbf{A}^{h_i}\mathbf{f}_i$, where $\mathbf{H}_{ij} := \mathbf{L}_{ij} - \lambda_{ij}\mathbf{A}^{h_i}(\mathbf{A} \odot \mathbf{K}_{ij})$. Finally, when it is varied as all indices $i \in \{1, 2, 3, \dots, m\}$, we have

$$\mathbf{Hu} + \mathbf{Xd} = \mathbf{Bf}, \tag{43}$$

where the parameters contained in (43), except $\mathbf{H}$, are defined as same as (27) in which change $\lceil \alpha_i \rceil$ into $h_i$ instead and

$$\mathbf{H} := \begin{bmatrix} \mathbf{L}_{11} - \lambda_{11}\mathbf{A}^{h_1}(\mathbf{A} \odot \mathbf{K}_{11}) & \mathbf{L}_{12} - \lambda_{12}\mathbf{A}^{h_1}(\mathbf{A} \odot \mathbf{K}_{12}) & \cdots & \mathbf{L}_{1m} - \lambda_{1m}\mathbf{A}^{h_1}(\mathbf{A} \odot \mathbf{K}_{1m}) \\ \mathbf{L}_{21} - \lambda_{21}\mathbf{A}^{h_2}(\mathbf{A} \odot \mathbf{K}_{21}) & \mathbf{L}_{22} - \lambda_{22}\mathbf{A}^{h_2}(\mathbf{A} \odot \mathbf{K}_{22}) & \cdots & \mathbf{L}_{2m} - \lambda_{2m}\mathbf{A}^{h_2}(\mathbf{A} \odot \mathbf{K}_{2m}) \\ \vdots & \vdots & \ddots & \vdots \\ \mathbf{L}_{m1} - \lambda_{m1}\mathbf{A}^{h_m}(\mathbf{A} \odot \mathbf{K}_{m1}) & \mathbf{L}_{m2} - \lambda_{m2}\mathbf{A}^{h_m}(\mathbf{A} \odot \mathbf{K}_{m2}) & \cdots & \mathbf{L}_{mm} - \lambda_{mm}\mathbf{A}^{h_m}(\mathbf{A} \odot \mathbf{K}_{mm}) \end{bmatrix}.$$

For the given initial conditions, we can perform as same processes as to obtain (28) and (29) in the previous section. We finally have $\mathbf{S}'(\mathbf{I}_m \otimes \mathbf{S}^{-1})\mathbf{u} = \mathbf{b}$, where the parameters are defined as in Section 3.2 by instead changing $\lceil \alpha_i \rceil$ to $h_i$. Note that, it has exactly $\sum_{i=0}^{m} h_i$ equations. In practice, we can solve the systems (43) and above condition, simultaneously, by constructing them to the linear system in a block matrix form

$$\begin{bmatrix} \mathbf{H} & \mathbf{X} \\ \mathbf{S}'(\mathbf{I}_m \otimes \mathbf{S}^{-1}) & \mathbf{0} \end{bmatrix} \begin{bmatrix} \mathbf{u} \\ \mathbf{d} \end{bmatrix} = \begin{bmatrix} \mathbf{Bf} \\ \mathbf{b} \end{bmatrix}, \tag{44}$$

where $\mathbf{0}$ is the $\sum_{i=1}^{m} h_i \times \sum_{i=1}^{m} h_i$ zero matrix and other parameters are defined as mentioned above. Note that the linear system (44) has a total of $mM + \sum_{i=1}^{m} h_i$ equations. Hence, by solving (44), we obtain the numerical solution $u_i(x)$ for the problem (2).

### 4.2. Experimental Examples for System of CIDEs

In this section, we apply the proposed numerical procedure to find approximate solutions of the systems of CIDEs (2). We numerically demonstrate three examples that are computed via MatLab program to verify the accuracy of our algorithm.

**Example 4** ([32]). *Consider the system of linear first order CIDEs over $x \in [0, 1]$*

$$u_1'(x) + u_2(x) = 1 + x + x^2 - \int_0^x (u_1(t) + u_2(t))dt,$$

$$u_2'(x) - u_1(x) = -1 - x - \int_0^x (u_1(t) - u_2(t))dt,$$

*subject to the initial conditions $u_1(0) = 1$ and $u_2(0) = -1$. Whose analytical solutions of this example are given by $u_1^*(x) = x + e^x$ and $u_2^*(x) = x - e^x$.*

This experiment is the system of first order CIDEs which is considered under constant coefficients, constant kernel functions and polynomial forcing terms. By using our numerical procedure, we obtain the approximate solutions $u_1(x)$ and $u_2(x)$ for each $x \in [0, 1]$. A comparison of the absolute error between $u_1(x)$ and $u_2(x)$ obtained from our proposed method, Genocchi polynomials method (GPM) [32] and biorthogonal system in approximation (BSA) [33], with their exact solutions by using $M = 8$ as shown in Table 7. The run-time is about 0.0437 s.

**Table 7.** A comparison of absolute errors of $u_1(x)$ and $u_2(x)$ for Example 4.

| $x$ | GPM [32] | | BSA [33] | | FIM-SCP | |
|---|---|---|---|---|---|---|
| | $Eu_1(x)$ | $Eu_2(x)$ | $Eu_1(x)$ | $Eu_2(x)$ | $Eu_1(x)$ | $Eu_2(x)$ |
| 0.2 | $1.1926 \times 10^{-8}$ | $7.5681 \times 10^{-9}$ | $4.9477 \times 10^{-8}$ | $3.4781 \times 10^{-6}$ | $9.0571 \times 10^{-10}$ | $4.5561 \times 10^{-10}$ |
| 0.6 | $1.2158 \times 10^{-8}$ | $3.7151 \times 10^{-9}$ | $8.9823 \times 10^{-7}$ | $3.7114 \times 10^{-5}$ | $1.7562 \times 10^{-9}$ | $7.7135 \times 10^{-10}$ |
| 1.0 | $2.5729 \times 10^{-8}$ | $1.9506 \times 10^{-8}$ | $1.5028 \times 10^{-5}$ | $1.2451 \times 10^{-4}$ | $6.3656 \times 10^{-10}$ | $3.5808 \times 10^{-10}$ |

**Example 5** ([34]). *Consider the system of linear second order CIDEs over $x \in [0,1]$*

$$u_1'' + (-3x^2 + 6x - 7)u_1 + (-2x^3 - 2x^2)u_2 = f_1 + \int_0^x \left( \kappa_{11} u_1 + \kappa_{12} u_2 \right) dt,$$

$$u_2'' + 2(x - 1)u_1 + (2x^4 + 2x^3 + 2x^2 - 1)u_2 = f_2 + \int_0^x \left( \kappa_{21} u_1 + \kappa_{22} u_2 \right) dt,$$

*where $f_1 = x^4 - x^3 - 2x^2 - 6$, $f_2 = x^4 + 3x^2 - 2$, $\kappa_{11} = t^3 - x^3$, $\kappa_{12} = (xt)^2 - x^4$, $\kappa_{21} = x^2 - t^2$ and $\kappa_{22} = -(xt)^2 - x^4$ with $u_1(0) = u_2(0) = u_1'(0) = 1$ and $u_2'(0) = -1$. The analytical solutions are $u_1^*(x) = e^x$ and $u_2^*(x) = e^{-x}$.*

This example is a system of linear second order CIDEs with variable coefficients, polynomial forcing terms and kernel functions are in term of functions depending on variables $x$ and $t$. By hiring our procedure with $M = 8$, we obtain the numerical solutions $u_1(x)$ and $u_2(x)$ for each $x \in [0,1]$. We compare the absolute errors between our algorithm and STWS [34] as displayed in Table 8. The run-time is 0.0880 s.

**Table 8.** A comparison of absolute errors of $u_1(x)$ and $u_2(x)$ for Example 5.

| $x$ | STWS [34] | | FIM-SCP | |
|---|---|---|---|---|
| | $Eu_1(x)$ | $Eu_2(x)$ | $Eu_1(x)$ | $Eu_2(x)$ |
| 0.2 | $8.59 \times 10^{-7}$ | $2.25 \times 10^{-7}$ | $1.4956 \times 10^{-10}$ | $2.6680 \times 10^{-10}$ |
| 0.6 | $4.70 \times 10^{-6}$ | $1.06 \times 10^{-6}$ | $5.0321 \times 10^{-10}$ | $5.3544 \times 10^{-10}$ |
| 1.0 | $1.11 \times 10^{-5}$ | $4.69 \times 10^{-6}$ | $4.4410 \times 10^{-15}$ | $2.3993 \times 10^{-15}$ |

From Examples 4 and 5, we can see that our proposed method for solving the system of CIDEs provides higher accuracy than other existing methods in terms of the absolute errors at the same number of nodes and under the same conditions which can be seen in Tables 7 and 8. Their average run-times are also consumed very inexpensive. In addition, we demonstrate the Euclidean error norm and the convergence orders $p$ or $\mathcal{O}(M^{-p})$ of the obtained solutions from Examples 4 and 5 in Table 9 for varying nodes $M$. We can see that our proposed procedure acquires a significant improvement in term of accuracy with less computational nodes $M$. This table also shows that the convergence orders rapidly increase when the number of nodes $M$ ever increases.

**Table 9.** Convergence orders of the approximate solutions for Examples 4 and 5.

| $M$ | Example 4 | | Example 5 | |
|---|---|---|---|---|
| | $\|\mathbf{u}_M^* - \mathbf{u}_M\|_2$ | **Order $p$** | $\|\mathbf{u}_M^* - \mathbf{u}_M\|_2$ | **Order $p$** |
| 4 | $1.7668 \times 10^{-3}$ | - | $3.2209 \times 10^{-2}$ | - |
| 5 | $9.5749 \times 10^{-5}$ | 13.064 | $3.0215 \times 10^{-3}$ | 10.605 |
| 6 | $4.3033 \times 10^{-6}$ | 17.016 | $2.1610 \times 10^{-4}$ | 14.468 |
| 7 | $1.6451 \times 10^{-7}$ | 21.175 | $1.0621 \times 10^{-5}$ | 19.545 |

Actually, we can also apply the devised numerical procedure for solving system of CIDEs to overcome the stiff system of ODEs by vanishing the kernel functions $\kappa_{ij}$ or the parameters $\lambda_{ij}$. Thus, we further illustrate two examples of the stiff systems.

**Example 6.** *Consider the following stiff system of linear first order ODEs over $x \in [0,1]$.*

$$u_1'(x) = 98u_1(x) + 198u_2(x),$$
$$u_2'(x) = -99u_1(x) - 199u_2(x),$$

*subject to the initial conditions $u_1(0) = u_2(0) = 1$. The analytical solutions are $u_1^*(x) = 4e^{-x} - 3e^{-100x}$ and $u_2^*(x) = -2e^{-x} + 3e^{-100x}$.*

By employing the proposed algorithm, we obtain the approximate solutions $u_1(x)$ and $u_2(x)$ at different $x$ close to zero as shown in Table 10. When they are compared with the Runge-Kutta fourth-order method (RK4) at $M = 16$, we have that our obtained results give higher accuracy. We also capture the solutions with $M = 30$ near zero as depicted in Figure 4a,b. Moreover, we found that the RK4 method is unavailable for solving this problem whole domain $[0,1]$ when small discretization. In contrast, our procedure can treat this trouble as illustrated in Figure 4c,d with $M = 50$. It is shown that these found solutions perfectly match with the exact.

**Table 10.** Absolute errors of $u_1(x)$ and $u_2(x)$ at different $x$ close to zero for Example 6.

| $x$ | RK4 | | FIM-SCP | |
|---|---|---|---|---|
| | $Eu_1(x)$ | $Eu_2(x)$ | $Eu_1(x)$ | $Eu_2(x)$ |
| 0.0025 | $8.5486 \times 10^{-8}$ | $8.5486 \times 10^{-8}$ | $2.3999 \times 10^{-15}$ | $1.1102 \times 10^{-15}$ |
| 0.0050 | $1.3998 \times 10^{-7}$ | $1.3998 \times 10^{-7}$ | $3.2335 \times 10^{-15}$ | $7.7716 \times 10^{-16}$ |
| 0.0075 | $1.8766 \times 10^{-7}$ | $1.8766 \times 10^{-7}$ | $3.8858 \times 10^{-15}$ | $9.9920 \times 10^{-16}$ |
| 0.0100 | $1.9206 \times 10^{-7}$ | $1.9206 \times 10^{-7}$ | $3.3584 \times 10^{-15}$ | $7.7716 \times 10^{-16}$ |

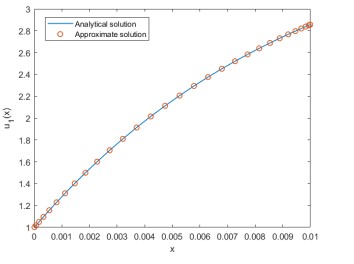
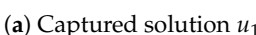
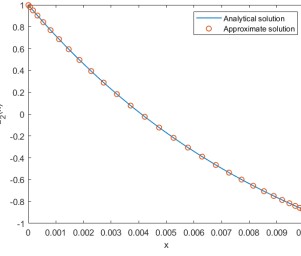
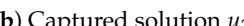
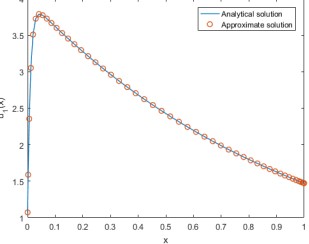
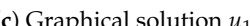
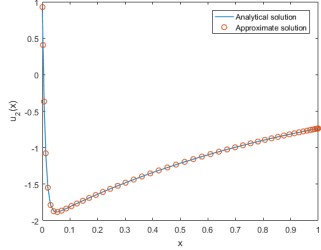

(**a**) Captured solution $u_1$    (**b**) Captured solution $u_2$    (**c**) Graphical solution $u_1$    (**d**) Graphical solution $u_2$

**Figure 4.** The graph of the approximate and exact solutions in Example 6.

## 5. Conclusions

In this article, the FIM-SCP is implemented to devise two numerical procedures for solving the systems of linear FIDEs and CIDEs. For the system of FIDEs (1), the fractional derivative is considered in the Caputo sense which is manipulated by the novel operational matrix of fractional integration (SCFM) as shown in Theorem 3. Then, the first procedure is devised by combining FIM-SCP and SCFM to overcome the systems of FIDEs (1) as expressed by the linear system (30). Next, the second procedure is created to deal with the generalized system of CIDEs (2) based on the FIM-SCP as expressed by the linear system (44). Note that both proposed numerical procedures are based on the FIM-SCP which completely eliminate the dilemma in the well-known round-off and discretization errors. These procedures are also in form of the linear systems. Thus, it is very convenient to find approximate solutions, we just substitute parameters of a given problem and

supplementary conditions into those linear systems and solve them by MatLab solver. Moreover, we can see from several Examples 1–5 that the devised algorithms provide much higher accuracy than other methods, consume low computational cost in terms of CPU time(s) and acquire a significant improvement in terms of absolute error with less computational nodes $M$. The obtained convergence orders for each example also give highly order. In addition, the second approach can solve the stiff system of linear ODEs as shown in Example 6. The obtained solutions are in good agreement with the exact solution. An interesting direction for our future work is to extend our techniques to solve the multi-dimensional FIDEs and CIDEs.

**Author Contributions:** Conceptualization, A.D., R.B. and M.J.; methodology, R.B. and A.D.; software, A.D. and M.J.; validation, A.D., R.B. and M.J.; formal analysis, R.B.; investigation, A.D. and M.J.; writing—original draft preparation, A.D. and M.J.; writing—review and editing, R.B.; visualization, A.D. and M.J.; supervision, R.B.; project administration, R.B.; funding acquisition, R.B. All authors have read and agreed to the published version of the manuscript.

**Funding:** The first author is supported by the Second Century Fund (C2F) for Postdoctoral Fellowship, Chulalongkorn University.

**Institutional Review Board Statement:** Not applicable.

**Informed Consent Statement:** Not applicable.

**Data Availability Statement:** Not applicable.

**Conflicts of Interest:** The authors declare no conflict of interest.

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
