# Peer review of "Numerical Solutions for Systems of Fractional and Classical Integro-Differential Equations via Finite Integration Method Based on Shifted Chebyshev Polynomials"

_fractalfract, doi:10.3390/fractalfract5030103_

Round 1
Reviewer 1 Report
Overall, the paper is well prepared and organized.
It can be accepted after the polishing the abstract, making it more logically and compact.
Author Response
Thank you for the comments. Please see our response in the file attached.

Reviewer 2 Report
In the paper, the finite integration method with shifted Chebyshev expansion is implemented to devise two numerical procedures for solving the systems of linear FIDEs and CIDEs, respectively. To demonstrate the efficiency, accuracy and convergence order of the procedures, several experimental examples are given. The proposed algorithm is also compared with numerical examples in published papers to prove its effectiveness. The research work is of certain significance and may be revised and published.
- The length of the paper is too long to concentrate on this problem. Numerical methods of fractional differential equations or integer differential equations can be discussed.
- There are some definitions and properties in the paper that are not used in the paper and do not need to be listed in the paper.
- What is the computation amount of the proposed algorithm? For example, formula (30) of the equations formed, the amount of calculation to solve the equations should be relatively large, please explain.
- It is recommended that the final revision of the paper should not exceed 20 pages.
Author Response

(The authors gave the same response as above.)

Reviewer 3 Report
In this work, authors proposed a fractional order-based method (Caputo sense) based on shifted Chebyshev polynomial and applied it to solve related problems of Systems of Fractional and Classical Integro-Differential Equations by constructing the corresponding operational matrices of integration. They also applied it to solve the stiff system of ODEs. The authors also presented some numerical examples to demonstrate the validity, applicability and accuracy of the proposed Shifted Chebyshev wavelet method.
This paper is acceptable and would be greatly enhanced with the following improvements.
A) The authors claimed the stability of the method, it is not clear how this been achieved. To get more well theoretical result, they should provide at least an upper bound and/or comparison results with those in the literature that used different wavelets. Can the authors include a small paragraph discussing in which way this novel method is more convenient than others (computational cost, accuracy)?
B) The paper contains too many examples compared to the theoretical achievements. I suggest reducing some of these examples
C) Proofs of Theorem 2,3 are not clear. More detailed steps need to be analyzed to the reader.
D) The sentences from line 210-216 need to be clarified. How you got this Order?
E) Fractional calculus has become one of the most studied field used to explain physical properties of the real-world problems such as COVID, SIR, health problems. Therefore, authors may consider citing the below published papers to enhance their manuscript:
https://doi.org/10.1007/s40096-019-0276-6
https://doi.org/10.1186/s13662-021-03262-7
Author Response

(The authors gave the same response as above.)
